# Multiagent Q-learning with Sub-Team Coordination

**Wenhan Huang**[1]*, **Kai Li**[2], **Kun Shao**[2], **Tianze Zhou**[3], **Matthew E. Taylor**[4,5], **Jun Luo**[2], **Dongge Wang**[6], **Hangyu Mao**[2], **Jianye Hao**[2,7], **Jun Wang**[8], **Xiaotie Deng**[9]†

[1]Shanghai Jiao Tong University, [2]Huawei Noah's Ark Lab, [3]Beijing Institute of Technology
[4]University of Alberta, [5]Alberta Machine Intelligence Institute (Amii), [6]EPFL
[7]Tianjin University, [8]University College London, [9]Peking University
rowdark@sjtu.edu.cn, {likai210,shaokun2,jun.luo1,maohangyu1}@huawei.com
simsimiztz@126.com, matthew.e.taylor@ualberta.ca, dongge.wang@epfl.ch
jianye.hao@tju.edu.cn, jun.wang@cs.ucl.ac.uk, xiaotie@pku.edu.cn

## Abstract

In many real-world cooperative multiagent reinforcement learning (MARL) tasks, teams of agents can rehearse together before deployment, but then communication constraints may force individual agents to execute independently when deployed. Centralized training and decentralized execution (CTDE) is increasingly popular in recent years, focusing mainly on this setting. In the value-based MARL branch, credit assignment mechanism is typically used to factorize the team reward into each individual's reward — individual-global-max (IGM) is a condition on the factorization ensuring that agents' action choices coincide with team's optimal joint action. However, current architectures fail to consider local coordination within sub-teams that should be exploited for more effective factorization, leading to faster learning. We propose a novel value factorization framework, called *multiagent Q-learning with sub-team coordination* (QSCAN), to flexibly represent sub-team coordination while honoring the IGM condition. QSCAN encompasses the full spectrum of sub-team coordination according to sub-team size, ranging from the monotonic value function class to the entire IGM function class, with familiar methods such as QMIX and QPLEX located at the respective extremes of the spectrum. Experimental results show that QSCAN's performance dominates state-of-the-art methods in matrix games, predator-prey tasks, the Switch challenge in MA-Gym. Additionally, QSCAN achieves comparable performances to those methods in a selection of StarCraft II micro-management tasks.

## 1 Introduction

For many complex real-world tasks, such as robot swarms [1] and autonomous vehicles [2], the coordination of agent teams is critical. Cooperative multiagent reinforcement learning (MARL) is a common setting where teams of agents, potentially using deep learning, can learn to solve a common task. This paper considers the centralized training with decentralized execution (CTDE) [3] framework, where a team of agents can train together with communication but execute fully decentralized during deployment (e.g., because of high communication costs, the risk of being detected).

Value-based methods in the CTDE paradigm have become increasingly popular as more capable methods have been proposed. These methods typically factorize the global action-value function

---

*This work was done when Wenhan Huang was an intern at Huawei Noah's Ark Lab.
†Corresponding author

36th Conference on Neural Information Processing Systems (NeurIPS 2022).

into agents' individual action-value functions to deal with the exponential growth of the joint-action space in the number of agents. If the solution to the problem can be identified so that the individual-global-max (IGM) property [4] can be satisfied, the problem can be simplified — in this case, an individuals' best (greedy) action corresponds to the best action for the team. Rather than having to reason about the full optimal joint action, the IGM condition allows individual agents to maximize their own individual action values.

Among these value-based methods, various factorization architectures satisfying the IGM condition have been proposed. Sunehag et al. [5] propose a linear factorization of the global action-value function named VDN. VDN treats the global action-value function as the sum of each agent's action value. QMIX [6] extends this factorization to a monotonic function so that QMIX can represent some non-linear global action-value functions. Qatten [7] decomposes the global action-value function with a multi-head attention approach based on the Taylor-expansion at the point of the optimal joint strategy, which finds more precise monotonic functions than QMIX. However, these structures cannot represent non-monotonic global action-value functions, which might cause the *relative overgeneralization* [8, 9] problem: The reward of one agent's cooperative action might be underestimated due to other agents' non-cooperative behavior.

To extract coordination information in tasks requiring non-monotonic global action-value functions, Deep Coordination Graph (DCG) [8] factorizes the global action-value function into pairwise action-value functions and converts the original problem to a co-ordination polymatrix game [10], leading to violation of the IGM condition. Due to the violation, DCG requires several rounds of inner-agent message passing at execution time. Wang et al. [11] propose another approach, QPLEX, which exploits the advantage-based IGM condition and achieves a complete IGM function class. However, this method treats different joint actions as totally different atomic items, which may lead to sample inefficiency and poor generalization.

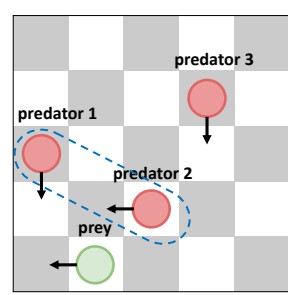

Figure 1: Sub-team coordination. In this simple predator-prey game, three predators (red circles) try to catch a prey (green circle). At this moment, only the sub-team of predators 1 and 2 matters because predator 3 is too far away.

In many settings, some agents must coordinate for success, but full coordination of all agents is not required. Instead, the task can be divided into several sub-tasks, and each sub-task requires exactly one sub-team of agents [12]. As Fig 1 shows, the sub-team of predator 1 and predator 2 is enough to catch the prey. The sub-team structure has been widely used for robotic tasks and unmanned aerial vehicle tasks with communication [13]. Yang et al. [14] also show that the coordination among sub-teams is enough for solving many complicated cooperative multiagent tasks under the CTDE paradigm.

Inspired by the sub-team organization, we study the sub-team factorization of the global action value and propose a hierarchical structure, called *multiagent Q-learning with Sub-team CoordinAtioN* (QSCAN), to strike a tradeoff between the representation capability of coordination patterns and the complexity of function classes. Our model generalizes the monotonic and the IGM structures via the sub-team factorization of the global action value. QSCAN guarantees the IGM condition and flexibly handles sub-team formations. Two approaches, QPAIR and QSCAN, are proposed for our hierarchical structure QSCAN. We empirically evaluate these methods in several coordination tasks, including a matrix game, predator-prey tasks [15], and the Switch challenge [16]. Comparing our approaches with QMIX and QPLEX, we illustrate the sub-team coordination pattern improves the results in these tasks. QSCAN significantly outperforms the two baselines in these three settings, while QPAIR achieves comparable performance. In addition, we show that our approaches are also comparable with these baselines in some widely used StarCraft Multi-Agent Challenge (SMAC) scenarios [17]. These results show that our method can not only significantly benefit CTDE learning, but also suggest a way forward for more flexible sub-team coordination and learning in multiple settings beyond the CTDE.

## 2 Background

**Dec-POMDP.** We characterize a fully cooperative multiagent leaning task as a *decentralized partially observable Markov decision process* (Dec-POMDP) with a tuple $\mathcal{M} = \langle \mathcal{N}, \mathcal{S}, \mathcal{A}, T, \Omega, O, r, \gamma \rangle$ [18]. $\mathcal{N} = \{1, 2, \ldots, n\}$ is a set of agents and $\mathcal{S}$ is a set consisting of environment states. At each time step, each agent $i \in \mathcal{N}$ chooses an action $a_i \in \mathcal{A}_i$, forming a joint action $\boldsymbol{a} \in \mathcal{A} = \times_{i=1}^n \mathcal{A}_i$ of all agents. This leads to a transition from the current state $s$ to the next one $s'$ governed by a transition function $T(s'|s, \boldsymbol{a})$. Due to the partial observability, $\Omega = \times_{i=1}^n \Omega_i$ is the joint observation set, where $\Omega_i$ is the partial observation set of agent $i$. $O(\boldsymbol{o}|s, \boldsymbol{a}')$ is the conditional probability of joint observations given the current state $s$ and the previous joint action $\boldsymbol{a}'$. $r : \mathcal{S} \times \mathcal{A} \to \mathbb{R}$ denotes the global reward function. The objective of the task is to maximize the total discounted reward $\sum_{t=0}^{\infty} \gamma^t r(s_t, \boldsymbol{a}_t)$ where $\gamma \in [0, 1)$ is a discount factor, $s_t$ and $\boldsymbol{a}_t$ are the state and the joint action at time step $t$ respectively.

**Multiagent deep Q-learning.** For a given joint policy $\boldsymbol{\pi}$ of agents, let the global action-value function $Q^{\boldsymbol{\pi}}(s_t, \boldsymbol{a}_t) = \mathbb{E}[\sum_{k=0}^{\infty} \gamma^i r(s_{t+k}, \boldsymbol{a}_{t+k})]$ denotes the expected discounted reward starting at state $s_t$ with joint action $\boldsymbol{a}_t$. When $\boldsymbol{\pi}$ is optimal, the global action-value function satisfies the Bellman optimality equation. Due to the partial observation setting, we would use the joint observation history $\boldsymbol{\tau}$ in place of the state $s$. Multiagent deep Q-learning represents the global action-value function $Q_{tot}$ with a deep neural network parameterized by $\boldsymbol{\theta}$. In centralized algorithms, the parameters $\boldsymbol{\theta}$ are obtained by minimizing the expected TD error $\mathcal{L} = \mathbb{E}\left[\left(r + \max_{\boldsymbol{a}'} Q_{tot}(\boldsymbol{\tau}', \boldsymbol{a}'; \boldsymbol{\theta}^-) - Q_{tot}(\boldsymbol{\tau}, \boldsymbol{a}; \boldsymbol{\theta})\right)^2\right]$, where $\boldsymbol{\theta}^-$ is the parameter of a target network.

**Value-based methods in the CTDE paradigm.** A typical dilemma in cooperative multiagent learning is that each agent should act independently based on its observation, but global information is needed to make a good decision for teamwork. CTDE [3] is the popular framework these years to remedy this challenge. In most value-based methods in the CTDE paradigm, the individual action-value functions for all players are trained in a centralized way, and an agent only relies on its trained individual function and its partial observation to choose actions. Due to the partial observation, [4] suggests that the optimal joint strategy should be executed as each agent plays its own optimal strategy concurrently. Formally, the condition of such action-value functions can be described as follows:

**Definition 1 (Individual-Global-Max (IGM) [4])** *For a global action-value function $Q_{tot} : \mathcal{T} \times \mathcal{A} \to \mathbb{R}$ and a series of individual action-value functions $[Q_i]_{i=1}^n$ with $Q_i : \mathcal{T}_i \times \mathcal{A}_i \to \mathbb{R}$, where $\boldsymbol{\tau} \in \mathcal{T}$ is the joint history and $\tau_i \in \mathcal{T}_i$ is agent $i$'s individual history, if $\forall \boldsymbol{\tau} \in \mathcal{T}$*

$$\left( \underset{a_1 \in \mathcal{A}_1}{\arg\max} \, Q_1(\tau_1, a_1), \ldots, \underset{a_n \in \mathcal{A}_n}{\arg\max} \, Q_n(\tau_n, a_n) \right) \in \underset{\boldsymbol{a} \in \mathcal{A}}{\arg\max} \, Q_{tot}(\boldsymbol{\tau}, \boldsymbol{a}),$$

*then we say that $[Q_i]_{i=1}^n$ satisfy the IGM condition for $Q_{tot}$.*

Once a framework satisfies the IGM condition, the global action value can be maximized efficiently via individually optimal choices of all the agents.

**Self-attention mechanism.** The self-attention mechanism [19] is widely used to extract the relationship between different positions of an input sequence in the natural language processing community. It can efficiently relate its different inputs. In the MARL community, the self-attention mechanism is used to learn the relationship between a team of agents. Jiang and Lu [20] employ the self-attention mechanism to learn the communication among agents. Li et al. [21] use the self-attention mechanism to determine the implicit coordination graph structure of agents. The formula of the self-attention mechanism can be written as: $\text{Attention}(\mathcal{Q}, \mathcal{K}, \mathcal{V}) = \text{softmax}(\mathcal{Q}\mathcal{K}^T)\mathcal{V}$, where $\mathcal{Q}$, $\mathcal{K}$, $\mathcal{V}$ denote the query vectors, the key vectors, and the value vectors, respectively. The weight of each value $\mathcal{V}_j$ to the $i$-th output is computed by the compatibility $\mathcal{Q}_i \mathcal{K}_j$, a dot product of $i$-th query vector and $j$-th key vector. The softmax activation function translates the dot product into a measurement of the attention.

## 3 Sub-Team Coordination Patterns

In a large multiagent system, the task is usually decomposed as several sub-tasks so that agents can accomplish each sub-task separately. For each sub-task, a sub-team is formed with several agents.

Besides, one agent could simultaneously belong to different sub-teams. This sub-team organization is general and can implicitly characterize most coordination patterns among agents [12]. In this paper, we exploit sub-team coordination patterns in value factorization frameworks. To this end, we would first explore sub-team coordination patterns in this section and then discuss the concrete architectures in the next one. The organization of this section is as follows. First, we propose a factorization of global action value based on the sub-team organization, named *sub-team factorization*. Then, we consider two typical coordination patterns, individuals as sub-teams and the grand team, used in previous works. Finally, we present a general framework characterizing coordination patterns with different sub-team sizes. For simplicity, we only consider the fully observable settings in this section.

**Sub-team factorization.** With the sub-team organization, each sub-team is assigned to a sub-task. Ideally, the sub-tasks are disjoint so that the agents beyond a sub-team would not affect the outcome of this sub-team. Mathematically, each sub-team $\boldsymbol{ST} \subseteq \mathcal{N}$ has an action-value function $Q_{\boldsymbol{ST}}(s, \boldsymbol{a}_{\boldsymbol{ST}})$ for the performance of its sub-task and we have $\forall j \notin \boldsymbol{ST}$, $\frac{\partial Q_{\boldsymbol{ST}}}{\partial Q_j} = 0$. $\boldsymbol{a}_{\boldsymbol{ST}}$ is the joint action of agents in $\boldsymbol{ST}$. The global action value $Q_{tot}$ can be treated as an evaluation of the set of sub-teams' action values $\{Q_{\boldsymbol{ST}}\}$. Since the sub-tasks are disjoint, $Q_{tot}$ should be monotonic with each $Q_{\boldsymbol{ST}}$, that is $\frac{\partial Q_{tot}}{\partial Q_{\boldsymbol{ST}}} \geq 0$.

**Individuals.** A typical solution is to treat each agent $i$ as a sub-team $\{i\}$ so that each agent is assigned with a distinct sub-task. In this scenario, we can assume $Q_{\{i\}} = Q_i$ and $\frac{\partial Q_{tot}}{\partial Q_i} \geq 0$. For a given state $s$, the function class can be rewritten as $Q_{tot}(s, \boldsymbol{a}) = f(Q_1(s, a_1), \ldots, Q_n(s, a_n), s)$, where $f$ is a monotonic mixing function. It is identical to the monotonic value factorization used by QMIX [22]. As in [4], this formulation immediately satisfies the IGM condition. However, this monotonic factorization may limit the representation capacity of the framework and lead to incorrect solutions during the training process [4, 15].

One may consider encoding more coordination patterns into the same value factorization framework but with a non-monotonic function $f$. However, without any constraints for the input of $f$, we prove in Proposition 1 (detailed proof in Appendix B.4) that the considered function space satisfying the IGM condition should be equivalent to the space in which the monotonic factorization can represent. In this proposition, we assume the input of $f$ can take arbitrary values because the framework should satisfy the IGM condition *consistently during the course of training*.

**Proposition 1** *Consider a fixed mixing function $f : \mathbb{R}^n \times \mathcal{S} \to \mathbb{R}$. If $Q_{tot}(s, \cdot) = f(Q_1, \ldots, Q_n, s)$, where $Q_{tot}$ and $[Q_i]_{i=1}^n$ satisfy the IGM condition consistently for any function $Q_i(s, \cdot)$ which contains a unique maximum point, then $f$ should satisfy $\forall i \, \forall x_i \in \mathbb{R}$, $\frac{\partial f(x_1, \ldots, x_n, s)}{\partial x_i} \geq 0$.*

**The grand team.** Only considering the grand team $\{1, \ldots, n\}$ is another typical solution. In this factorization, we can assume $Q_{\{1, \ldots, n\}} = Q_{tot}$. With the individual action values $\{Q_i\}$, the global action value can be written as $Q_{tot}(s, \boldsymbol{a}) = f(Q_1, \ldots, Q_n, s, \boldsymbol{a})$. This kind of value factorization may not always satisfy the IGM condition.

QPLEX [11], an instance of this function class, employs the duplex dueling structure and transfers the IGM condition to advantage values $A(s, \boldsymbol{a}) = Q(s, \boldsymbol{a}) - V(s)$, where $V(s) = \max_{\boldsymbol{a}} Q(s, \boldsymbol{a})$. The global advantage value $A_{tot}$ is factorized as $A_{tot}(s, \boldsymbol{a}) = \sum_{i=1}^n \lambda_i(s, \boldsymbol{a}) A_i(s, a_i)$, where $[\lambda_i(s, \boldsymbol{a})]_{i=1}^n$ is an importance weight. The joint action $\boldsymbol{a}$ here is considered atomic. As pointed in [11], the IGM function space is equivalent to the space that QPLEX can represent. However, the indivisibility of the joint action $\boldsymbol{a}$ in the grand team may prevent the mixing network from reusing the knowledge from previous coordination patterns, leading to poor generalization. Specifically, the importance weights using atomic joint actions in QPLEX

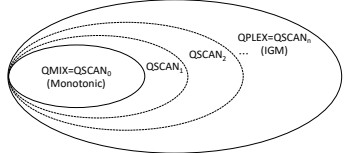

Figure 2: Coordination hierarchy. We identify and prove a general representation of relation among sub-team coordination function classes, from the monotonic function class to the IGM function class.

may not perform well in credit assignment when the correlations among agents become complicated. As the predator-prey results shown in [15] and [11], QPLEX requires additional tuning of the exploration rate to extract predators' coordination patterns.

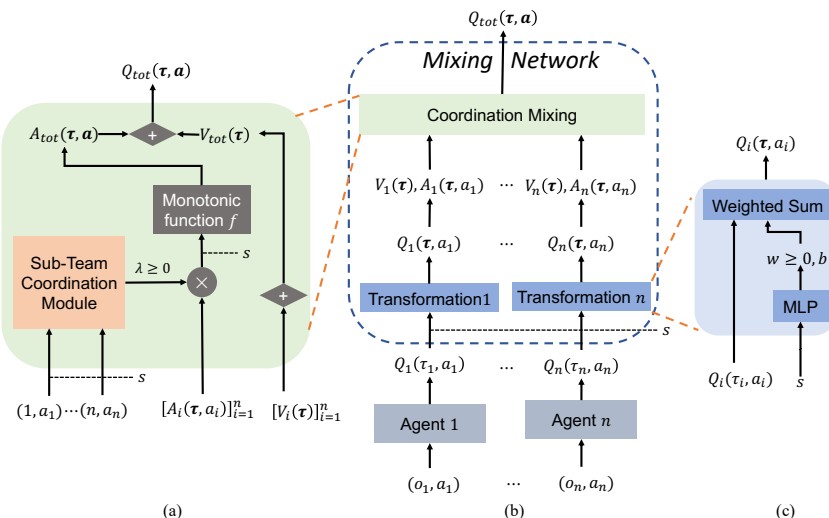

Figure 3: The base architecture of QSCAN. (a) Coordination mixing sub-network structure. (b) The overall QSCAN architecture. (c) Transformation module.

**$k$-member sub-teams.** To provide a general view of the sub-team organization and characterize coordination patterns more systematically, we consider the organization with $k$-member sub-teams, where $k \in \{1, 2, \ldots, n\}$. We have been pursuing a value factorization framework $\text{QSCAN}_k$, which could characterize the coordination patterns within the $k$-member sub-teams. To this end, we would first analyze the properties of $\text{QSCAN}_k$ and then discuss the concrete architecture in the next section.

$\text{QSCAN}_k$ should characterize the coordination patterns beyond $k$-member sub-teams, and thus we would accordingly establish a hierarchical structure of these QSCAN classes. To this end, we first show the relation between $\text{QSCAN}_k$ and $\text{QSCAN}_{k-1}$ in some special team reward functions. For several types of team reward, including $c$-degree polynomial functions, $c$-clause functions, and hypergraphical functions [23], coordination patterns within $(k-1)$-member sub-teams can be transferred to coordination patterns within $k$-member sub-teams (see Appendix B.2). More generally, when the sub-team size is typically small as in practice ($k \leq \frac{n}{2}$), according to Hall's marriage theorem [24], $\text{QSCAN}_{k-1}$ can be shown as a special case of $\text{QSCAN}_k$ with the only assumption $\frac{\partial Q_{tot}}{\partial Q_{ST'}} \geq 0$ (see Appendix B.3). Although, for more general cases, such relationship might no longer exist when $k > \frac{n}{2}$ (except for $k = n$, when the sub-team is exact the grand team), the hierarchical structure of coordination patterns inspires us on the design of our value factorization method. As the special team reward functions above, intuitively, a linear or multiplicative factorization based on some sub-team features can ensure the hierarchical property. We pursue that our QSCAN establishes a hierarchical structure for all sub-team sizes, named coordination hierarchy, shown in Fig 2.

## 4 Multiagent Q-learning with Sub-team Coordination

In this section, we present the concrete architecture of QSCAN, which is a value factorization framework exploiting sub-team coordination patterns while honoring the IGM condition. We first describe the base architecture, which employs duplex dueling structures [11] to guarantee the IGM condition. We then discuss the crucial component of the QSCAN, sub-team coordination module, which characterizes the organization and coordination patterns of the team. A series of function classes could be derived from the QSCAN framework. Finally, we discuss the coordination module in detail and present two practical architectures for QSCAN.

**Base architecture.** The overall architecture of our model is shown in Fig 3. The duplex dueling structure factorizes each agent $i$'s action-value function into its individual value and an advantage function as $V_i(\boldsymbol{\tau}) = \max_{a'_i} Q_i(\boldsymbol{\tau}, a'_i), A_i(\boldsymbol{\tau}, a_i) = Q_i(\boldsymbol{\tau}, a_i) - V_i(\boldsymbol{\tau})$. The global action-value function is factorized similarly as $Q_{tot}(\boldsymbol{\tau}, \boldsymbol{a}) = V_{tot}(\boldsymbol{\tau}) + A_{tot}(\boldsymbol{\tau}, \boldsymbol{a})$, where the global value $V(\boldsymbol{\tau}) = \sum_{i=1}^{n} V_i(\tau)$. The global advantage function is represented as $A_{tot}(\boldsymbol{\tau}, \boldsymbol{a}) = f([\lambda_i(\boldsymbol{\tau}, \boldsymbol{a})A_i(\boldsymbol{\tau}, a_i)]_{i=1}^{n}, s)$, where

non-negative weights $\lambda$ is generated through a *Sub-team Coordination* module and $f$ is a monotonic function with respect to each $x_i \leq 0$ (i.e., $\frac{\partial f}{\partial x_i} \geq 0$), and maintains a maximum value $0$ ($f(\mathbf{0}_n, s) = 0$). It can be shown that this structure satisfies the IGM condition (see Proposition 2 in Appendix B.5). Following previous works [6, 11, 25], we would use the global state $s$ as the centralized information, if applicable, or the joint history $\boldsymbol{\tau}$.

**Sub-team coordination module and QSCAN framework.** We analyze the design of the theoretical coordination module which characterizes coordination patterns within $k$-member sub-teams and then propose the QSCAN framework. Intuitively, the team reward can be credited to each sub-team and then to each individual. Consider a sub-team $\boldsymbol{ST}$ containing $k$ agents and $\boldsymbol{a_{ST}}$ is $\boldsymbol{ST}$'s joint action. The contribution of $\boldsymbol{ST}$ can be assigned to each member $i$ with an importance weight $g_i^{\boldsymbol{ST}}$ evaluating $i$'s contribution in $\boldsymbol{ST}$. Therefore, we could approximately factorize global advantage into each agent's individual advantage

$$
\begin{aligned}
A_{tot}(\boldsymbol{\tau}, \boldsymbol{a}) &\approx \sum_{\boldsymbol{ST}:\boldsymbol{ST}\subseteq\mathcal{N},|\boldsymbol{ST}|=k} \left[ \sum_{i\in\boldsymbol{ST}} \left( g_i^{\boldsymbol{ST}}(\boldsymbol{\tau}, \boldsymbol{a_{ST}}) \cdot A_i(\boldsymbol{\tau}, a_i) \right) \right] \\
&= \sum_{i=1}^{n} \left( \sum_{\boldsymbol{ST}:i\in\boldsymbol{ST}\subseteq\mathcal{N},|\boldsymbol{ST}|=k} g_i^{\boldsymbol{ST}}(\boldsymbol{\tau}, \boldsymbol{a_{ST}}) \right) A_i(\boldsymbol{\tau}, a_i).
\end{aligned}
$$

Based on this factorization, we propose QSCAN$_k$.

**Definition 2 (QSCAN$_k$)** *QSCAN$_k$ is a branch of QSCAN which concerns coordination among sub-teams containing only $k$ agents. Specifically, QSCAN$_k$ adopts the following weights yield from the sub-team coordination module $\lambda_i(\boldsymbol{\tau}, \boldsymbol{a}) = h\left( \sum_{\boldsymbol{ST}:i\in\boldsymbol{ST}\subseteq\mathcal{N},|\boldsymbol{ST}|=k} g_i^{\boldsymbol{ST}}(\boldsymbol{\tau}, \boldsymbol{a_{ST}}) \right)$, where $h$ is a non-negative activation function.*

$\lambda_i(\boldsymbol{\tau}, \boldsymbol{a})$ corresponds to the total contribution of agent $i$ to all sub-teams containing $k$ members based on joint history $\boldsymbol{\tau}$. Since the advantage as the disparity from the optimal action should keep non-positive, $h$ is used to ensure positivity of each $\lambda$. According to QSCAN$_k$, we could obtain QSCAN$_0$ by the analytic continuation, in which $g_i^{\boldsymbol{ST}}$ does not take actions as input. With the continuation, Fig 2 illustrates that QMIX and QPLEX locate at the respective extremes of the spectrum for our QSCAN framework. The detailed explanation and proofs for these propositions are in Appendix B.7 and B.8.

After demonstrating the QSCAN framework, we present two different coordination architectures: `QPAIR` is based on the natural enumeration of sub-teams, and `QSCAN` is based on the self-attention mechanism.

**Pairwise coordination.** We first present a pairwise coordination module of `QPAIR` by enumerating all sub-teams with size $2$. The module uses a multi-layer perceptron (MLP) to calculate the pairwise coordination coefficients (Fig 4a). This structure is a QSCAN$_2$ level in the coordination hierarchy. The coefficient of each $A_i(\tau_i, a_i)$ is based on pairs of agent's action as well as the global state $(s, i, a_i, j, a_j)$. Then $\lambda_i(\boldsymbol{\tau}, \boldsymbol{a}) = h\left( \sum_{j=1}^{n} \mathrm{MLP}(s, i, a_i, j, a_j) \right)$. In practice, we use absolute value function as $h$ here.

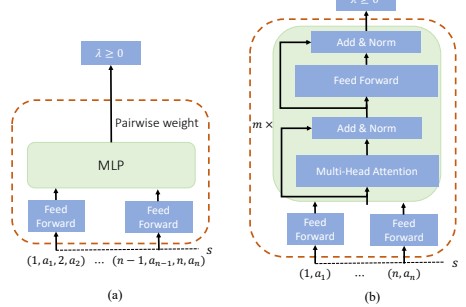

Figure 4: Coordination modules. (a) Pairwise coordination module. (b) Self-attention module.

The simple extension of `QPAIR` for larger sub-team size $k$ increases the computation cost exponentially due to the enumeration of $O(n^k)$ sub-teams. However, such exponential enumeration might not be required because identifying the optimal sub-team coordination patterns is enough for the task. That is, the problem becomes searching the optimal patterns, which can be solved via data driven methods. This kind of searching-optimal problems have been approximately solved by neural

network structures. For example, some combinatorial optimization problems like traveling salesman problem or Boolean satisfiability have been approximated by some neural network structures [26, 27].

**Self-attention.** Inspired by the structure of Transformer [19], we may use self-attention mechanisms to characterize correlations among agents hierarchically. The self-attention mechanism has been shown to be universal approximators for continuous permutation equivariant functions [28], where the importance weights $g_i^{ST}$ and $\lambda_i$ belong to. Meanwhile, the self-attention mechanism can aggregates agent-action information effectively and flexibly. Fig 4b illustrates the sketch of our module of QSCAN. The module takes all agent-action pairs $[(i, a_i)]_{i=1}^n$ and the global state $s$ as input and produces a series of non-negative weights $[\lambda_i]_{i=1}^n$.

Let us explain how the agent-action information is aggregated through the self-attention mechanism. The embedded input vector of agent-action pairs $[(i, a_i)]_{i=1}^n$ can be viewed as the individual agent-action information vector or the action information vector of 1-member sub-teams. When the action information vector of $l$-member sub-teams passes an attention layer, the pairwise weights $\mathcal{Q}_i \mathcal{K}_j$ will relate the $i$-th and $j$-th positions of the vector. Notice that the action information of $2l$-member sub-team can be obtained by aggregating the action information of the first $l$ members' sub-team and that of the last $l$ members' sub-team. Therefore, the output of this attention layer can be viewed as the action information vector of $2l$-member sub-teams. Without considering the residue links in Fig. 4b, the output of the $m$-th attention layer compresses the action information of $2^m$-member sub-teams. The residue links will mix up the action information of sub-teams with different sizes, and it provides the network more flexibility to learn different forms of the coordination patterns.

# 5 Empirical Results

We compare QPAIR and QSCAN with state-of-the-art MARL approaches, QMIX and QPLEX, in various coordination tasks, including matrix games, predator-prey challenges [8], the Switch task [16], and the StarCraft Multi-Agent Challenge (SMAC) [17]. For a fair comparison, the implementation of QSCAN uses one attention layer, that is, with sub-teams of size 2. The learning curves are plotted with a smooth factor of 0.6 except the last point. The implementation detail and environment detail are in Appendix C. In the matrix game, the agents need explicit coordination to obtain the highest payoff. The predator-prey tasks are complicated scenarios with immediate coordination rewards. In these tasks, the agents need to learn spatial-temporal local coordination. The Switch task is a more complicated coordination task due to the sparse and long-term rewards. In this task, the sub-team coordination in the beginning steps influences intensely over the final rewards. The SMAC environment is common-used for evaluating the MARL approaches. We evaluate our approaches in a selection of SMAC scenarios. Additional empirical results including ablation study for the number of attention layers of QSCAN and a super-hard scenario 27m_vs_30m are in Appendix E.

## 5.1 Matrix Game

Table 1 shows the payoffs of a 3-player with 2-action matrix game. Fig 5a illustrates the empirical results for QMIX, QPLEX, QPAIR and QSCAN. Due to the relative overgeneralization, QMIX fails to find the optimal solution. QPLEX suffers another problem about the poor generalization. QSCAN and QPAIR can find the optimal solution more quickly because the pairwise coordination patterns provide more suitable generalization in this task. In the exploration period, the pairwise coordination would allow the algorithm to explore the optimal solution more easily. Moreover, for all different random seeds in our experiments, QSCAN always finds the optimal solution rapidly in this matrix game.

Table 1: Payoffs of a 3-player matrix game. Each player $i \in \{1, 2, 3\}$ has two actions $\{A, B\}$. The complete payoff matrix is split into two sub-matrices according to player 1's action $a_1$.

When $a_1$=A

| $a_2$ \ $a_3$ | A | B |
|---|---|---|
| A | 0 | 0 |
| B | **23** | 12 |

When $a_1$=B

| $a_2$ \ $a_3$ | A | B |
|---|---|---|
| A | 17 | 20 |
| B | 0 | 17 |

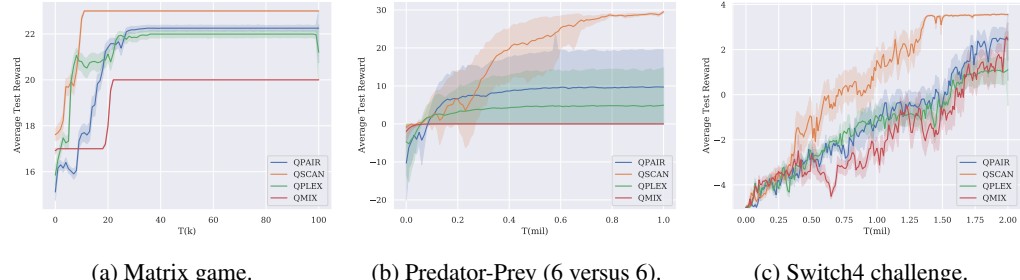

| (a) Matrix game. | (b) Predator-Prey (6 versus 6). | (c) Switch4 challenge. |

Figure 5: Learning curves of `QPAIR`, `QSCAN`, QPLEX, and QMIX in three different tasks. For the timestamp $T$, k and mil are short for is 'kilo' and 'million', respectively. The average reward with 95% confidence intervals is shown. Best viewed in color.

## 5.2 Predator-Prey

We compare our approaches `QPAIR`, `QSCAN` with QMIX and QPLEX in partially-observable cooperative *Predator-Prey* tasks created by [8]. We evaluate algorithms in the scenario with 6 predators against 6 prey.

The results are shown in Fig 5b. QMIX fails to learn a positive reward due to the relative over-generalization caused by the miscoordination of "capture" actions. QPLEX can find some correct coordination among agents. For `QPAIR`, it performs better than QPLEX because the correct coordination of "capture" actions only needs pairwise coordination, which `QPAIR` is forced to learn. For `QSCAN`, it outperforms all these approaches due to its adaptive balance of the pairwise coordination and each individual's local information.

Fig 6a illustrates the `QSCAN`'s self-attention weights for continuous 3 steps over the complete process that predator 2 and predator 5 capture a prey. The figures on the left side demonstrate the state, where a predator is described as a rectangle with agent id on it, and the preys are denoted as the yellow rectangles. The figures on the right side show the corresponding self-attention heat maps. For each map, a rectangle lying in row $i$ and column $j$ represents the agent $i$'s attention for agent $j$. Brighter the color is, larger the weight is, and more attention is attracted. In the first step of the capture process, all predators pay attention to predator 2 who attempts to capture the prey in row 4 and column 3. In the second step, predator 2 pays attention to predator 5 who responses, and they successfully coordinate to capture the prey in row 4 and column 2. When current capture is completed, predator 2 and predator 5 finish their jobs and predator 1 will organize the next capture process.

## 5.3 Switch4 in MA-Gym

Switch4 in MA-Gym [16] is a partially observable task that 4 agents need to reach their corresponding home by passing through the one-agent wide narrow corridor as Fig 7.

As Fig 5c shows, our approach `QSCAN` outperforms others in this task while QPLEX performs worst. QMIX and `QPAIR` achieve comparable results, while our approach achieves better performance during the training phase.

We show the self-attention heat map of `QSCAN` in the first 3 steps of optimal and sub-optimal solutions respectively. Fig 6b shows the heat map for both optimal and sub-optimal solutions. The left side of Fig 6b is the heat map of the optimal solution. In the first step, all agents pay attention to agent 2 who will lead the first passing. In the

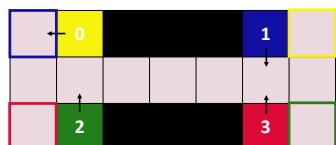

Figure 7: The starting state of Switch4. 4 agents need to reach their corresponding destinations. Each colored block (yellow, blue, green, or red) with a number denotes an agent. Black blocks are barriers. Blocks with colored boundaries denote the destinations for corresponding colored agents.

mean time, agent 2 pays attention to agent 1 because agent 1 should not move down to cause a traffic jam. In the second step, agent 2 pays attention to agent 0 who needs to follow agent 2 to pass the corridor. In the third step, each agent follows the coordination organized in the first two steps. The right side of Fig 6b is the heat map of the optimal solution. Similarly, in the first step, all agents pay

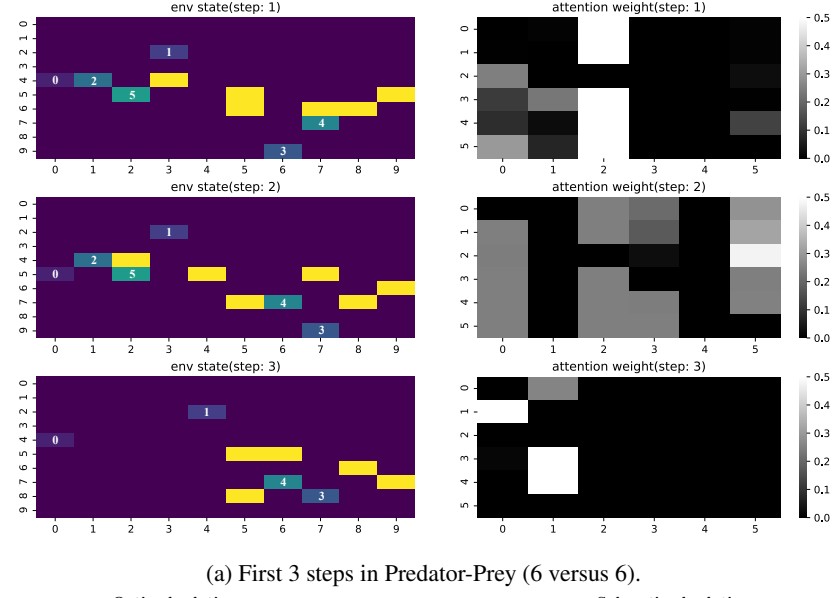

(a) First 3 steps in Predator-Prey (6 versus 6).

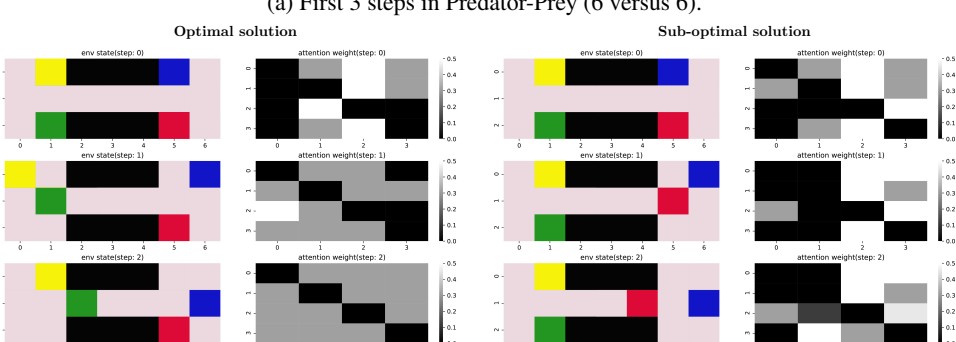

(b) First 3 steps in different solutions for Switch4.

Figure 6: States of the games and the corresponding attention heat-maps. In each of 3 columns of pictures, the left side shows the states and the right side shows the attention. (a) In these states, the predators are numbered from 0 to 5 and the preys are yellow rectangles. (b) We demonstrate two solutions. See Fig 7 for more details of the states.

attention to agent 2, while agent 2 pays attention to agent 3 and asks agent 3 to pass the corridor first. In the second step, all agents pay attention to both agent 2 and agent 3 that they will organize the passing. In the third step, agent 3 additionally pays attention to agent 1, who needs to follow agent 3 to pass the corridor.

## 5.4 The StarCraft Multi-Agent Challenge

The StarCraft Multi-Agent Challenge (SMAC) [17] is a widely-used benchmark for cooperative MARL. We evaluate our approaches on a wide-range of SMAC scenarios, including homogeneous (e.g., 5m_vs_6m) and heterogeneous (e.g., 3s5z) agents. Furthermore, we compare our approaches with state-of-the-art baselines: QMIX and QPLEX. The empirical results are shown in Table 2, and the corresponding figures are presented in Appendix E.4.

Table 2: Median of the test win rates in the SMAC.

| Scenario | QSCAN | QPAIR | QPLEX | QMIX |
|----------|-------|-------|-------|------|
| 2s_vs_1sc | **100** | **100** | **100** | **100** |
| 2s3z | **100** | **100** | **100** | 97 |
| 3s5z | **98** | 97 | 97 | 94 |
| 1c3s5z | 88 | **97** | **97** | 94 |
| 2c_vs_64zg | 62 | 86 | **88** | 42 |
| 5m_vs_6m | 76 | **77** | 72 | 69 |

The results shows that `QSCAN` and `QPAIR` are superior to QMIX in most scenarios. `QSCAN` performs almost well except for 2c_vs_64zg. Its win rate is about 20 percents less than that of `QPAIR` or QPLEX. 2c_vs_64zg needs the coordination of two colossi, where the pairwise coordination characterized by `QPAIR` coincides with the grand team coordination characterized by QPLEX. As QMIX does not perform well in this scenario, we conjecture that this scenario requires the pairwise coordination patterns rather than individual patterns. One possible reason for the performance of `QSCAN` could be that the individual patterns through the residue links prevent `QSCAN` to achieve better performance in this scenario. `QPAIR` achieves the best performance in some scenarios and no more than 2 percents less than the best one's win rate in other selected scenarios. Overall, our approaches achieve comparable performances with the SOTA baseline QPLEX which uses the joint action in the mixing function.

## 6 Conclusions and Future Work

We propose QSCAN, a novel value-based multiagent reinforcement learning framework that can characterize coordination patterns within sub-teams hierarchically and guarantee the IGM condition. QSCAN provides value factorization architectures with an expressive mixing network for the centralized end-to-end training and learns a series of individual action-value functions for decentralized execution. We establish a coordination hierarchy based on QSCAN after analyzing the sub-team factorization, and present two efficient implementations based on pairwise coordination and self-attention mechanisms. Empirically, we show that our methods achieve better or comparable performance with baselines in several benchmarks.

While our value-based architecture employs the duplex dueling structure for the sub-team factorization, we believe that the sub-team factorization benefits other architectures, e.g., policy-based methods like COMA [25]. As our discussion about sub-teams in Sec. 3, designing more effective hierarchical structures for sub-team organization beyond the factorization on advantages, remains a challenge. From theoretical side, it remains a problem about whether a general coordination hierarchy exists according to the sub-team factorization $\frac{\partial Q_{tot}}{\partial Q_{ST}} \geq 0$ when sub-team size $k \geq \frac{n}{2}$. One more thing, there has not been a universal organizational paradigm suitable for most tasks [29, 30]. Besides the sub-team organization, exploring other organizational paradigms in cooperative MARL is promising. In the future, we will continue exploring organizational paradigms in large multiagent systems.

## Acknowledgments and Disclosure of Funding

We would like to thank Weixun Wang, Xiaotian Hao, and Zipeng Dai for various helpful discussions. This work is partially supported by the National Natural Science Foundation of China (Grant No. 62172012). Part of this work has taken place in the Intelligent Robot Learning (IRL) Lab at the University of Alberta, which is supported in part by research grants from the Alberta Machine intelligence Institute; a Canada CIFAR AI Chair, Amii; CIFAR; Compute Canada; and NSERC.

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
