# Appendix for
# Multiagent Q-learning with Sub-Team Coordination

## Contents

# A  Related Work

**Value factorization architectures.** Decentralizing the agents' joint policy has long been a central issue of the value-based cooperative MARL in the popular CTDE paradigm [1]. It allows each agent has an individual action-value function while the whole team's global action-value function could be trained in a centralized way. Sunehag et al. [2] introduced a linear action-value factorization architecture, called value decomposition network (VDN), whereby the global action value is just a sum of the individual action values. The greedy joint policy could thus be obtained by maximizing each agent's individual action-value function simultaneously. QMIX [3] improves this linear architecture by introducing a monotonic mixing network, which connects the global action value with the individual action values. The monotonic function class increases the representational capacity of value factorization architectures. Son et al. [4] analyzed the properties of previous factorization frameworks and proposed a general condition on the factorization, called individual-global-max (IGM), ensuring the agents' individually optimal actions are consistent with the optimal joint actions. QPLEX [5] leverages the duplex dueling structure $Q = V + A$ [6] to achieve the complete expressiveness of the IGM function class. The duplex dueling structure allows the architecture to maintain the largest capacity of representation in the IGM function class.

**Related training schemes.** In addition to identifying suitable value factorization *architectures*, various *training schemes* have been proposed to enhance the applicability of the structures. For instance, WQMIX [7] uses a projection operator to project the actual global action-value function into QMIX function space. In this way, WQMIX transfers the challenging task into an easily factorizable one that QMIX could factorize. Similar conclusions about QTRAN [4] can be drawn. We regard these methods as training schemes because other architectures could also use the same projection operators to train. Since we are interested in value factorization architectures in this paper, we did not compare our methods with the structures exploiting these training schemes.

**Organizational paradigms.** The organization of a multiagent system is the structure of relationships that govern the agents' behavior. Substantial evidence suggests that employing domain-dependent organizational structures could significantly affect the performance of the system [8]. Various organizational paradigms, including coalitions and teams, have emerged and been applied to numerous domains. A coalition formed by a subset of agents can be treated as a single atomic entity, whose utility is the summation of individual agents' utilities. Several MARL approaches using coalitions and Shapley values [9–11] aim to derive agents' marginal contribution to the grand team for achieving a "fair" credit assignment. A team of agents [12, 13] would coordinate with each other toward a common goal. Within a team, the coordination patterns could be various, and generally, each agent could belong to different sub-teams. In contrast to the competitive "coalitions", all "sub-teams" have a common goal that is to maximize the utility of the grand team. Typically, an agent should exactly belong to one coalition, while the agent can belong to several sub-teams concurrently. Numerous teamwork structures have emerged in previous works [12, 14]. In this paper, rather than deriving the marginal contributions of the agents via coalitions, we employ sub-team structures to characterize coordination patterns among cooperative agents.

**Relationship to VAST [11].** VAST is a multiagent actor-critic learning algorithm, which employs the time-variable coalition as its organization and uses a centralized critic to factorize the global value into individual values for training actors. At one particular time step, VAST divides agents into several groups where each agent belongs to exactly one group (same as one coalition) at one particular time step. In contrast, an agent might belong to different groups at different time steps. After dividing, a coalition value is the sum of the individual values of its agents, and the coalition values pass the centralized critic to obtain the global value. In the VAST paper [11], VAST employs the mixing function of QMIX or QTRAN as its centralized critic. When focusing on one time step, we can view VAST's factorization as a subclass of QMIX's monotonic factorization (as QTRAN projects the actual global action-value function into the additive function). In this viewpoint, the VAST algorithm allows the input order of agents' individual values to vary over time rather than introducing a new factorization method.

**Graph neural networks and self-attention in cooperative MARL.** In cooperative MARL, Graph Neural Networks (GNNs, [15]) are typically used to characterize the dependence of graphs by message passing between the nodes in the graphs. Self-attention [16] is an attention mechanism that could be used to compute a representation of a sequence of entities. Therefore, both GNNs and

self-attention could be used to characterize the global information of the team in the cooperative MARL, which is conditioned on the observations and the actions of the agents. Previous works in MARL use GNNs and self-attention mechanisms to extract neighboring agents' features from the individual side [17–19], or build a centralized critic or a mixing network from the team side [20–22]. Since we are interested in value factorization architectures, we focus on related works that use GNNs and self-attention as centralized critics or mixing networks. In contrast to our implementation of `QSCAN` which employs self-attention mechanisms to characterize *coordination patterns*, previous works [20–22] only consider employing GNNs and self-attention to extract features of the whole team for a representation of the global action-value function.

## B  Omitted Proofs and Discussions

### B.1  Discussions about Monotonic Value Function Factorization

In the mainbody of our work, we point out that the monotonic value function factorization like QMIX may limit the representation capacity of the global action-value function. An immediately interesting issue arises that if we care only about the *optimal* global action-value function, whether this monotonic factorization could have enough representation capacity to characterize the optimal solution. We will show in the following that this is true and the optimal global action-value function of the task can always be factorized by some monotonic function. However, this kind of solution may not be practical because it is pretty hard for the architecture to learn the solution.

We consider a fully observable setting where each agent's observations are the full state in this subsection. This setting is equivalent to a multiagent Markov decision process (MMDP) [23]. We focus on the mixing function $f$ which takes individual action values and the global state as input, i.e., $Q_{tot}(s, \boldsymbol{a}) = f(Q_1(s, a_1), \ldots, Q_n(s, a_n), s)$. We show that for any MMDP, there exists a mixing function and a series of individual action-value functions, such that the output of the mixing function satisfies the Bellman optimality equation. In addition, we show that the output of this mixing function and those individual action-value functions satisfy the IGM condition. For convenience, we let $V^* : \mathcal{S} \to \mathbb{R}$ denote the optimal state value function for an MMDP.

**Lemma 1.** *For any MMDP, there exists a mixing function $f : \mathbb{R}^n \times \mathcal{S} \to \mathbb{R}$ and a series of individual action-value functions $[Q_i]_{i=1}^n$ where $Q_i : \mathcal{S} \times \mathcal{A} \to \mathbb{R}$ contains a unique maximum point, such that $Q_{tot}(s, \boldsymbol{a}) = f(Q_1(s, a_1), \ldots, Q_n(s, a_n), s)$ with $\boldsymbol{a} = (a_1, \ldots, a_n)$ satisfies the Bellman optimality equation*

$$\forall s \in \mathcal{S}, \quad V^*(s) = \max_{\boldsymbol{a}} Q_{tot}(s, \boldsymbol{a}), \tag{1}$$

*in the meantime $[Q_i]_{i=1}^n$ satisfy the IGM condition for $Q_{tot}$.*

*Proof.* Without loss of generality, we only consider a specific state $s$. Suppose that $\boldsymbol{a}^* = (a_1^*, \ldots, a_n^*)$ is the optimal joint action and $V^*(s)$ is the optimal state value for state $s$. For any $\epsilon > 0$, we can create a series of individual action-value functions $[Q_i]_{i=1}^n$ with

$$Q_i(s, a_i) = \begin{cases} \text{sgn}(V^*(s)) & \text{if } a_i = a_i^*, \\ \text{sgn}(V^*(s)) - \epsilon & \text{otherwise.} \end{cases}, \tag{2}$$

where $\text{sgn}(\cdot)$ is the sign function. Obviously, each $Q_i$ contains a unique maximum point $a_i^*$. We can then assemble a global action-value function $Q_{tot}$ by

$$Q_{tot}(s, \boldsymbol{a}) = \left[ \frac{1}{n} \sum_{i=1}^n Q_i(s, a_i) \right] |V^*(s)|. \tag{3}$$

It would be easy to verify that $[Q_i]_{i=1}^n$ satisfy IGM for $Q_{tot}$. Since for any agent $i$, $a_i^*$ is the maximum point of $Q_i(s, \cdot)$, we have

$$\forall \boldsymbol{a}, \quad Q_{tot}(s, \boldsymbol{a}) \leq Q_{tot}(s, \boldsymbol{a}^*) = \left[ \frac{1}{n} \sum_{i=1}^n \text{sgn}(V^*(s)) \right] |V^*(s)| = V^*(s), \tag{4}$$

which completes the proof. $\qquad\square$

**Remark 1.** *It should be noted that our construction of $Q_{tot}$ in Eq. (3) tries to match the true global action-value function over just* one *choice* of joint actions in each state, even if the true global function may have *multiple* optimal actions.*

## B.2 Discussion on Different Team Reward Functions

In this section, we will discuss several types of team reward: $c$-degree polynomial functions, $c$-clause functions, and hypergraphical functions. We will represent the formulation of team reward types and then show the existence of the hierarchical structures for each reward function.

First, we will discuss the $c$-degree polynomial functions. A $c$-degree polynomial function is a polynomial function with a degree at most $c$ for individual value functions, which can be used as a $c$-polynomial approximation for some other types of team reward. The mathematical formulation of a $c$-degree polynomial function is

$$Q_{tot} = \sum_{\boldsymbol{d}:\boldsymbol{d}\in\{0,1\}^n, \sum_{i=1}^n d_i \leq c} w_{\boldsymbol{d}} \prod_{i=1}^n Q_i^{d_i}.$$

It is trivial that a $c$-degree polynomial function can be represented by $c$-member sub-teams as

$$Q_{\boldsymbol{ST}} = \sum_{\boldsymbol{d}:\boldsymbol{d}\in\{0,1\}^n, \sum_{i=1}^n d_i \leq c} \left\{ \frac{1}{\binom{n}{c-\sum_{i=1}^n d_i}} w_{\boldsymbol{d}} \prod_{i=1}^n Q_i^{d_i} \cdot \prod_{i=1}^n \mathbf{1}[d_i = 0 \vee i \in \boldsymbol{ST}] \right\},$$
$$Q_{tot} = \sum_{|\boldsymbol{ST}|=c} Q_{\boldsymbol{ST}},$$

where $\boldsymbol{ST}$ is a $k$-member sub-team. Notice that a $c$-degree polynomial function is also a $(c+1)$-degree polynomial function so that it can be represented by $(c+1)$-member sub-teams, which shows a hierarchical structure.

Second, we move to $c$-clause functions. A $c$-clause function is a CNF-like Boolean function with each clause corresponding to at most $c$ individual values. This type of team reward indicates that the task is done by accomplishing all sub-tasks, each of which is represented by a clause. Specifically,

$$Q_{tot} = \prod_{1 \leq x_1 < x_2 < \cdots < x_{c'} n, c' \leq c} f_{\boldsymbol{x}}(Q_{x_1}, Q_{x_2}, \ldots, Q_{x_{c'}}),$$

where $f_{\boldsymbol{x}}$ is a Boolean function. Since all $f_{\boldsymbol{x}}$ is either 0 or 1, a $c$-clause function can be represented by $c$-member sub-teams as

$$Q_{\boldsymbol{ST}} = \prod_{\boldsymbol{x}:1 \leq |\boldsymbol{x}|=c' \leq c, x \subseteq \boldsymbol{ST}} f_{\boldsymbol{x}}(Q_{x_1}, Q_{x_2}, \ldots, Q_{x_{c'}}),$$
$$Q_{tot} = \prod_{|\boldsymbol{ST}|=c} Q_{\boldsymbol{ST}}.$$

Since $c$-clause functions can be viewed as a special case of $(c+1)$-clause functions, $(c+1)$-member sub-teams can represent those functions. Therefore, a hierarchical structure is built upon coefficient $c$.

Finally, we consider the hypergraphical functions. A hypergraphical function is a team reward function of a cooperative hypergraphical game [24], where the team reward is the summation of the utilities of several hyperedges. The hypergraphical function can be viewed as a graphical approximation of other team reward function. Formally, the team reward is

$$Q_{tot} = \sum_{e=(x_1, x_2, \ldots, x_{|e|}) \in \mathcal{E}} u_e(a_1, a_2, \ldots, a_{|e|}),$$

where $\mathcal{E}$ is the set of hyperedges and $u_e$ is the utility function of the hyperedge $e$. When each $e \in \mathcal{E}$ satisfying $|e| \leq k$, the hypergraphical function can be represented by $k$-member sub-teams as

$$Q_{\boldsymbol{ST}} = \sum_{e:e\in\mathcal{E}, e \subseteq \boldsymbol{ST}} \left[ \frac{1}{\binom{n}{k-\sum_{i=1}^n d_i}} u_e(a_1, a_2, \ldots, a_{|e|}) \right],$$
$$Q_{tot} = \sum_{|\boldsymbol{ST}|=k} Q_{\boldsymbol{ST}},$$

where $e \subseteq \boldsymbol{ST}$ means that each agent in hyperedge $e$ is also a member of sub-team $\boldsymbol{ST}$. The construction for a representation based on $(k+1)$-member sub-teams is similar, which means the coordination patterns in $k$-member sub-team can be viewed as those in $(k+1)$-member sub-team. A hierarchical structure based on the maximal size of hyperedges can be built up.

Although $c$-degree polynomial functions, $c$-clause functions, and hypergraphical functions are different approximations for other team reward functions, they all form a hierarchical structure with the sub-team representation. This fact indicates that the hierarchical structure for team reward functions with the sub-team representation could exist in several general cases.

### B.3  General Cases for Sub-team Coordination Patterns

In this section, we show that the coordination patterns within a team of $n$ agents form a natural hierarchical structure based on the size $k$ of sub-teams when $k \leq \frac{n}{2}$. This result is only based on the assumption of the monotonic sub-team factorization of $k$-member sub-teams.

**Theorem 1.** *Given a team of $n$ agents and a sub-team size $k$ ($1 \leq k \leq \frac{n}{2}$), consider a monotonic factorization for a set of $k$-member sub-teams $A_k$, s.t. $\forall \boldsymbol{ST} \in A_k : \frac{\partial Q_{tot}}{\partial Q_{ST}} \geq 0$ and $\forall \boldsymbol{ST} \notin A_k : \frac{\partial Q_{tot}}{\partial Q_{ST}} = 0$. Then there exists a set of $k+1$-member sub-teams $A_{k+1}$, s.t. $\forall \boldsymbol{ST'} \in A_{k+1} : \frac{\partial Q_{tot}}{\partial Q_{ST'}} \geq 0$ and $\forall \boldsymbol{ST'} \notin A_{k+1} : \frac{\partial Q_{tot}}{\partial Q_{ST'}} = 0$.*

*Proof.* The idea to prove the theorem is to map each $k$-member sub-team $\boldsymbol{ST}$ to a corresponding $(k+1)$-member sub-team $\boldsymbol{ST'}$ so that we can construct $Q_{ST'}(\cdot, \boldsymbol{a_{ST}}) \equiv Q_{ST}(\boldsymbol{a_{ST}})$. Notice that, when each member in $\boldsymbol{ST}$ is also a member in $\boldsymbol{ST'}$, that is, $\boldsymbol{ST'}$ is a super-set of $\boldsymbol{ST}$, $Q_{ST'}$ can be equivalent to $Q_{ST}$ by simply ignoring the information of the member $\boldsymbol{ST'} \setminus \boldsymbol{ST}$.

To construct the mapping, we will build up a bi-part graph of $k$-member and $(k+1)$-member sub-teams, and then show the existence of such mapping via Hall's marriage theorem Hall [25].

Consider a bi-part graph $G = (\mathcal{A}, \mathcal{B}, \mathcal{E})$, where $\mathcal{A}, \mathcal{B}$ are the sets of $k$-member sub-teams and $(k+1)$-member sub-teams, respectively. $\mathcal{E}$ is the set of edges, where $\forall (\boldsymbol{ST}, \boldsymbol{ST'}) \in \mathcal{E}$ if and only if $\boldsymbol{ST} \in \mathcal{A}$, $\boldsymbol{ST'} \subset \mathcal{B}$, and $\boldsymbol{ST} \subset \boldsymbol{ST'}$. The degree for each $\boldsymbol{ST} \in \mathcal{A}$ is $(n-k)$, while that for each $\boldsymbol{ST} \in \mathcal{B}$ is $(k)$. Since $k \leq \frac{n}{2}$, we have $n - k \geq k$. Then size of the neighbours for each $\mathcal{A'} \subseteq \mathcal{A}$ is at least

$$|\mathcal{A}| \cdot \frac{n-k}{k} \geq |\mathcal{A}|.$$

By Hall's marriage theorem, there exists a perfect matching in graph $G$ for $\mathcal{A}$ side. With this perfect matching, we can construct $A_{k+1}$ by $A_k$ and corresponding $Q$-functions $Q_{ST'}$. Since $Q_{ST'}(\cdot, \boldsymbol{a_{ST}}) \equiv Q_{ST}(\boldsymbol{a_{ST}})$, we still satisfy the condition $\frac{\partial Q_{tot}}{\partial Q_{ST'}} \geq 0$. For the sub-teams outside $A_{k+1}$, we can set their $Q$-functions as zeros. Then we have

$$\frac{\partial Q_{tot}}{\partial Q_{ST'}} \geq 0 \quad \forall \boldsymbol{ST'} \in A_{k+1},$$

and

$$\frac{\partial Q_{tot}}{\partial Q_{ST'}} = 0 \quad \forall \boldsymbol{ST'} \notin A_{k+1},$$

which concludes the proof.

$\square$

For the case that $k > \frac{n}{2}$, this proof is no longer applicable. However, we do not know whether the hierarchical structure remains with the same assumption $\frac{\partial Q_{tot}}{\partial Q_{ST}} \geq 0$ or some counterexamples exist. It might be an open problem.

### B.4  Proof of Proposition 1

**Proposition 1.** *Consider a fixed mixing function $f : \mathbb{R}^n \times \mathcal{S} \to \mathbb{R}$. If $Q_{tot}(s, \cdot) = f(Q_1, \ldots, Q_n, s)$, where $Q_{tot}$ and $[Q_i]_{i=1}^n$ satisfy the IGM condition consistently for any function $Q_i(s, \cdot)$ which contains a unique maximum point, then $f$ should satisfy $\forall i \, \forall x_i \in \mathbb{R}, \frac{\partial f(x_1, \ldots, x_n, s)}{\partial x_i} \geq 0$.*

We show that the mixing function $f$ satisfying those requirements should be monotonic.

Without loss of generality, we only consider a specific state $s \in \mathcal{S}$. For any agent $i$, let $a_i^*$ be the optimal action for its individual action-value function, specifically

$$a_i^* = \arg\max_{a_i \in \mathcal{A}_i} Q_i(s, a_i). \tag{5}$$

Denote the vector of these optimal actions by $\boldsymbol{a}^* = (a_1^*, \ldots, a_n^*)$. Due to the uniqueness of the optimal actions, we immediately have $Q_i(s, a_i^*) > Q_i(s, a_i)$ for any $i$ and $a_i \neq a_i^*$. Since $Q_i$ can be an arbitrary function, without loss of generality, for any $\epsilon > 0$ and some $a_i \neq a_i^*$, we could create a function $Q_i$ satisfying $Q_i(a_i^*) - Q_i(a_i) = \epsilon$. For convenience, we define a helper function $\tilde{h}_i$

$$\tilde{h}_i(x_i) = f(Q_1(s, a_1^*), \ldots, Q_{i-1}(s, a_{i-1}^*), x_i, Q_{i+1}(s, a_{i+1}^*), \ldots, Q_n(s, a_n^*), s). \tag{6}$$

According to the IGM condition, $\boldsymbol{a}^*$ is a maximum point of $Q_{tot}(s, \cdot)$. Therefore, we have $\tilde{h}_i(Q_i(s, a_i^*)) \geq \tilde{h}_i(Q_i(s, a_i))$. It follows that

$$\left. \frac{\partial f(x_1, \ldots, x_n, s)}{\partial x_i} \right|_{\forall i, \, x_i = Q_i(s, a_i^*)} \tag{7}$$

$$= \lim_{\epsilon \to 0} \frac{\tilde{h}_i(Q_i(s, a_i^*)) - \tilde{h}_i(Q_i(s, a_i^*) - \epsilon)}{\epsilon} \tag{8}$$

$$= \lim_{\epsilon \to 0} \frac{\tilde{h}_i(Q_i(s, a_i^*)) - \tilde{h}_i(Q_i(s, a_i))}{\epsilon} \tag{9}$$

$$\geq 0 \tag{10}$$

Since $Q_i$ can be arbitrary, the partial deviates should be non-negative in the whole domain, which completes the proof.

## B.5 Base Architecture and the IGM Condition

We show that our base architecture guarantees the IGM condition.

**Proposition 2.** *The base network guarantees the IGM condition.*

For any $(Q_{tot}, [Q_i]_{i=1}^n)$ that base network can represent and any histories $[\tau_i]_{i=1}^n$, we let

$$\boldsymbol{a}^* = (a_1^*, \ldots, a_n^*) = \left( \arg\max_{a_1 \in \mathcal{A}_1} Q_1(\tau_1, a_1), \ldots, \arg\max_{a_n \in \mathcal{A}_n} Q_n(\tau_n, a_n) \right) \tag{11}$$

denote a vector of individual optimal actions. Due to the positive transformations and the monotonic function $f$ in the mixing network, it is quite straightforward to obtain $\forall i, \forall a_i \neq a_i^*$,

$$Q_i(\tau_i, a_i^*) = \max_{a_i'} Q_i(\tau_i, a_i') \tag{12}$$

$$\Rightarrow (\text{Positive linear transformation } w > 0) \quad Q_i(\boldsymbol{\tau}, a_i^*) = \max_{a_i'} Q_i(\boldsymbol{\tau}, a_i') \tag{13}$$

$$\Rightarrow (A_i(\boldsymbol{\tau}, a_i) = Q_i(\boldsymbol{\tau}, a_i) - Q_i(\boldsymbol{\tau}, a_i^*)) \quad A_i(\boldsymbol{\tau}, a_i^*) = 0 \text{ and } A_i(\boldsymbol{\tau}, a_i) < 0 \tag{14}$$

$$\Rightarrow (\text{Positive weights } [\lambda_i]_{i=1}^n \text{ and the monotonic function } f) \quad A_{tot}(\boldsymbol{\tau}, \boldsymbol{a}^*) = 0 \text{ and } A_{tot}(\boldsymbol{\tau}, \boldsymbol{a}) < 0 \tag{15}$$

$$\Rightarrow (Q_{tot}(\boldsymbol{\tau}, \boldsymbol{a}) = A_{tot}(\boldsymbol{\tau}, \boldsymbol{a}) + V_{tot}(\boldsymbol{\tau})) \quad Q_{tot}(\boldsymbol{\tau}, \boldsymbol{a}^*) = \max_{\boldsymbol{a}'} Q_{tot}(\boldsymbol{\tau}, \boldsymbol{a}'). \tag{16}$$

Therefore, $\boldsymbol{a}^*$ is the optimal joint action of $Q_{tot}(\boldsymbol{\tau}, \cdot)$, which satisfies the Definition 1.

## B.6 Sub-Team Factorization for Advantage Functions

For a particular joint history $\boldsymbol{\tau}$, consider the global advantage $A_{tot} = Q_{tot} - V_{tot}$, an action value $Q_{\boldsymbol{ST}}$ of sub-team $\boldsymbol{ST}$, and a individual advantage $A_i = Q_i - V_i$ of agent $i$.

When other agents' policies are fixed, $Q_{ST}$ can be treated as a function of $Q_i$. Taking the first order approximation,

$$Q_{ST} \approx \tilde{g}_i^{ST} Q_i + \tilde{C}_i^{ST}$$

, where $\tilde{g}_i^{ST}$ and $\tilde{C}_i^{ST}$ are the coefficients for the first order approximation. For the individual advantage $A_i$,

$$Q_{ST} \approx \tilde{g}_i^{ST} A_i + (\tilde{C}_i^{ST} + \tilde{g}_i^{ST} V_i). \tag{17}$$

Then we will consider the relationship between $A_{tot}$ and $A_i$. With discussion about whether $Q_{tot} = V_{tot}$ or not, it is easy to show $\frac{\partial Q_{tot}}{\partial Q_{ST}} \geq 0 \implies \frac{\partial A_{tot}}{\partial Q_{ST}} \geq 0$. Recall that (17) shows that $Q_{ST}$ is monotonic with $\tilde{g}_i^{ST} A_i$. $A_{tot}$ is monotonic with the value vector $\{\tilde{g}_i^{ST'} A_i\}_{ST':i \in ST'}$. The value vector size depends on the number of different sub-teams and grows exponentially with the size of sub-teams. One approximation is to encode the value vector into a single value so that the single value remains monotonic with $A_{tot}$. A simplest solution is to take the summation of each elements in the value vector $\sum_{ST':i \in ST'} \left( \tilde{g}_i^{ST'} A_i \right)$. However, this might not characterize the monotonicity between the $A_{tot}$ and the value vector. An improvement is to take the weighted summation of each elements, that is,

$$\sum_{ST':i \in ST'} \left( \hat{g}_i^{ST'} A_i \right)$$

, where $\hat{g}_i^{ST'}$ is a multiplication of the summation weight and $\tilde{g}_i^{ST'}$. With this approximation, $A_{tot}$ is monotonic with $\sum_{ST':i \in ST'} (\hat{g}_i^{ST'} A_i)$ for agent $i$. We take the first order approximation of $A_{tot}$ and slightly expand the representation of the coefficient:

$$A_{tot} \approx \sum_i \left[ \sum_{ST':i \in ST'} (g_i^{ST'} A_i) \right], \tag{18}$$

where $g_i^{ST'}$ is a coefficient combining $\hat{g}_i^{ST'}$ and the first order approximation coefficient of $A_{tot}$. In the approximation (18), we derive the individual advantages from the global advantage.

### B.7 QMIX in Coordination Hierarchy

**Proposition 3.** *QMIX is equivalent to $QSCAN_0$ in the Dec-POMDP setting, while they are equivalent to $QSCAN_1$ in the fully observable settings.*

We write the function class represented by the QMIX architecture in partially observable settings as

$$\text{QMIX} = \left\{ Q_{tot} \mid Q_{tot}(\boldsymbol{\tau}, \boldsymbol{a}) = \tilde{f}(Q_1(\tau_1, a_1), \dots Q_n(\tau_n, a_n), \boldsymbol{\tau}), \frac{\partial Q_{tot}}{\partial Q_i} \geq 0, Q_i(\tau_i, a_i) \in \mathbb{R} \right\}. \tag{19}$$

Similarly, we use $QSCAN_k$ and QPLEX to denote the corresponding function classes respectively.

In this subsection, we show that in the partially observable environment (Dec-POMDP), QMIX=$QSCAN_0$; while in the fully observable environment (MMDP), QMIX=$QSCAN_0$=$QSCAN_1$.

#### B.7.1 Partially Observable Settings

We prove QMIX=$QSCAN_0$ in the partially observable environment.

**$QSCAN_0 \subseteq$ QMIX** From the architecture of QSCAN, we can see that $QSCAN_0$ assembles $Q_{tot}$ by

$$Q_{tot}(\boldsymbol{\tau}, \boldsymbol{a}) = f(\lambda_1(\boldsymbol{\tau}) A_1(\boldsymbol{\tau}, a_1), \dots, \lambda_n(\boldsymbol{\tau}) A_n(\boldsymbol{\tau}, a_n), \boldsymbol{\tau}) + \sum_{i=1}^n V_i(\boldsymbol{\tau}), \tag{20}$$

where $f$ is non-decreasing with respect to $A_i$. It is quite straightforward to have $\forall i, \forall a_i \neq a_i'$,

$$Q_i(\tau_i, a_i) < Q_i(\tau_i, a_i') \tag{21}$$

$$\Rightarrow \text{(Positive linear transformation } w > 0) \quad Q_i(\boldsymbol{\tau}, a_i) < Q_i(\boldsymbol{\tau}, a_i') \tag{22}$$

$$\Rightarrow (A_i(\boldsymbol{\tau}, a_i) = Q_i(\boldsymbol{\tau}, a_i) - Q_i(\boldsymbol{\tau}, a_i^*)) \quad A_i(\boldsymbol{\tau}, a_i) < A_i(\boldsymbol{\tau}, a_i') \tag{23}$$

$$\Rightarrow \text{(Positive } \lambda_i(\boldsymbol{\tau}) \text{ is identical for different actions)} \quad \lambda_i(\boldsymbol{\tau})A_i(\boldsymbol{\tau}, a_i) < \lambda_i(\boldsymbol{\tau})A_i(\boldsymbol{\tau}, a_i') \tag{24}$$

$$\Rightarrow \text{(Monotonic function } f) \quad Q_{tot}(\boldsymbol{\tau}, (\ldots, a_i, \ldots)) < Q_{tot}(\boldsymbol{\tau}, (\ldots, a_i', \ldots)). \tag{25}$$

Therefore,

$$\forall i, \quad \frac{\partial Q_{tot}}{\partial Q_i} \geq 0. \tag{26}$$

Thus, we have $\text{QSCAN}_0 \subseteq \text{QMIX}$.

**QMIX $\subseteq$ QSCAN$_0$** For any $Q_{tot}$ represented by QMIX, specifically

$$Q_{tot}(\boldsymbol{\tau}, \boldsymbol{a}) = \tilde{f}(Q_1(\tau_1, a_1), \ldots Q_n(\tau_n, a_n), \boldsymbol{\tau}), \tag{27}$$

we can reconstruct it via the QSCAN$_0$ architecture. Since $\tilde{f}$ is a monotonic function, we use $\tilde{f}^*(\boldsymbol{\tau})$ to denote its maximum value

$$\tilde{f}^*(\boldsymbol{\tau}) = \tilde{f}(\max_{a_1} Q_1(\tau_1, a_1), \ldots, \max_{a_n} Q_n(\tau_n, a_n), \boldsymbol{\tau}). \tag{28}$$

For each individual action-value function $Q_i$, we can derive the individual value function $V_i$ and the advantage function $A_i$ by

$$V_i(\tau_i) = \max_{a_i'} Q_i(\tau_i, a_i'), \quad A_i(\tau_i, a_i) = Q_i(\tau_i, a_i) - V_i(\tau_i). \tag{29}$$

According to Eq. (20) and the transformation module in the mixing function, QSCAN$_0$ actually assembles $Q_{tot}$ by

$$Q_{tot}(\boldsymbol{\tau}, \boldsymbol{a}) = f(\lambda_1(\boldsymbol{\tau})w_1(\boldsymbol{\tau})A(\tau_1, a_1), \ldots, \lambda_n(\boldsymbol{\tau})w_n(\boldsymbol{\tau})A(\tau_n, a_n), \boldsymbol{\tau}) + \sum_{i=1}^{n} w_i(\boldsymbol{\tau})V_i(\tau_i) + b_i(\boldsymbol{\tau}). \tag{30}$$

If we set the parameters as follows

$$\forall i, \, w_i(\boldsymbol{\tau}) = 1, \lambda_i(\boldsymbol{\tau}) = 1, b_i(\boldsymbol{\tau}) = -V_i(\tau_i) + \frac{1}{n}\tilde{f}^*(\boldsymbol{\tau}), \tag{31}$$

and build our monotonic function $f$ from the target function $\tilde{f}$ as

$$\forall i, \forall x_i \in (-\infty, 0], \quad f(x_1, \ldots, x_n, \boldsymbol{\tau}) = \tilde{f}(x_1 + V_1(\tau_1), \ldots, x_n + V_n(\tau_n), \boldsymbol{\tau}) - \tilde{f}^*(\boldsymbol{\tau}), \tag{32}$$

then we could rebuild $Q_{tot}$ in Eq. (27) from our QSCAN$_0$ architecture. It could be easy to verify that $f$ satisfies the constraints of QSCAN, $f(\boldsymbol{0}, \boldsymbol{\tau}) = 0$ and $\max_{\boldsymbol{x}} f(\boldsymbol{x}, \boldsymbol{\tau}) = 0$. Therefore, we obtain QMIX $\subseteq$ QSCAN$_0$.

Combining QMIX $\subseteq$ QSCAN$_0$ and QMIX $\supseteq$ QSCAN$_0$, we obtain QMIX=QSCAN$_0$ in the partially observable environment.

### B.7.2 Fully Observable Settings

For the fully observable environments, we can see that $Q_{tot}$ represented by QSCAN$_1$ is

$$Q_{tot}(s, \boldsymbol{a}) = f(\lambda_1(s, a_1)A_1(s, a_1), \ldots, \lambda_n(s, a_n)A_n(s, a_n), s) + \sum_{i=1}^{n} V_i(s) \tag{33}$$

$$= f(A_1(s, a_1), \ldots, A_n(s, a_n), s) + \sum_{i=1}^{n} V_i(s). \tag{34}$$

This $Q_{tot}$ can be characterized exactly by QSCAN$_0$. Therefore, in the fully observable scenarios, QSCAN$_1$ degenerates to QSCAN$_0$. As we have already proved that QMIX=QSCAN$_0$, we finally obtain QMIX=QSCAN$_0$=QSCAN$_1$ in the fully observable environment.

**Remark 2.** *(i) In the partially observable environment, $QMIX \subsetneq QSCAN_1$. (ii) In the fully observable environment, it should be noted that although $QSCAN_1$ is equivalent to $QSCAN_0$ in representing $Q_{tot}$, they are different in the space of $(Q_{tot}, [Q_i]_{i=1}^n)$ because the mixing function of $QSCAN_1$ can be non-monotonic by adjusting the weight $\lambda_i(\boldsymbol{\tau}, a_i)$.*

### B.8   QPLEX in Coordination Hierarchy

**Proposition 4.** *QPLEX is equivalent to $QSCAN_n$. Therefore, $QSCAN_n$ can represent the whole IGM function space.*

We show that QPLEX=$QSCAN_n$.

In the mixing network of $QSCAN_n$, we could implement the monotonic function $f$ with a simple sum operator and this is just QPLEX. Thus, QPLEX $\subseteq QSCAN_n$.

On the other hand, we could prove QPLEX $\supseteq QSCAN_n$ by the help of the IGM function space. Since QPLEX is proved to be a superset of the IGM function space [5] and we have proved that QSCAN guarantees the IGM condition in Proposition 2, we obtain QPLEX $\supseteq$ IGM function space $\supseteq QSCAN_n$.

## C   Experimental Setup

### C.1   Testing Domains

**Matrix game** is a prototype of all games. We can convert every one-shot game with discrete actions to a matrix game. We construct a matrix game where the optimal solution and the sub-optimal solution are a pair of opposite points. The slope is sharper near the optimal solution than that near the sub-optimal one. The improper coordination, e.g. isolated individuals or the whole team, may lead to the sub-optimal solution rather than the optimal one. This game shows the importance of sub-team coordination.

**Predator prey** is a cooperative environment proposed by [26]. It is a coordination task that unco-operative actions cause punishment. The predators coordinate to capture the prey while they take a punishment when only one predator captures the prey. This task forces at least two predators to coordinate to capture the prey. In this paper, we set the punishment as $-0.5$ and consider 3 different settings shown in Fig 2: 6 vs 6, 8 vs 8, and 10 vs 10.

**Switch4** is a cooperative task in MA-Gym [27]. It is a coordination task with several different settings. In this paper, we consider the default partial observation task addressed as Switch4-v0. Each agent wants to move to its corresponding home in this task while it can only observe its position. The agents coordinate to take turns to pass the narrow corridor. We address the details of the environment parameters in Table 1.

**SMAC** [28] is a common-used environment to evaluate current state-of-art MARL approaches. In the SMAC environment, the enemy units are controlled by a built-in AI and the reinforcement learning agents need to defeat all enemy units by controlling each ally unit individually. The units in ally and enemy groups may be asymmetric. At each time step, each agent chooses one action for each unit from the discrete action space consisting of `move[direction]`, `attack[enemy_id]`, `stop`, and `no-op`. After that, a global reward is received by the MARL agents, which is calculated according to the damage point, the enemy unit kills, and a bonus for winning.

Table 1: Environment parameters of Switch4 game

| Environment parameter | Value | Description |
|---|---|---|
| `n_agent` | 4 | Number of agents |
| `n_action` | 5 | Left, right, up, down, stay |
| `max_step` | 50 | Max time step in a game |
| `full_observation` | False | Partial or full observation |
| `step_cost` | $-0.1$ | Additional reward for each step |
| `reward_done` | 1.25 | Reward for an agent reaching its home |

## C.2 Implementation Details

We adopt PyMARL [28] to run all experiments. The implementations and hyper-parameters of QMIX [3], QTRAN [4], and QPLEX [5] are the same as what they referred in their papers and their source codes. The hyper-parameters of our approaches used for Predator-Prey[26], MA-Gym [27], and SMAC [28] are the same. As Table 2 shows, our approach `QPAIR` uses similar hyper-parameters as QPLEX [5]. Our approach `QSCAN` employees a 1-layer self-attention with hyper-parameters in Table 3 implemented with PyTorch [29], while other common hyper-parameters are the same as those in `QPAIR`.

**General hyper-parameters**    We use the $\epsilon$-greedy exploration process in [7] for our approaches and all baselines. The default hyper-parameters are shown in Table 4.

## D  The StarCraft Multi-Agent Challenge

### D.1  Task Description

In the experiments, We evaluate our methods on a wide-range of SMAC scenarios, including homogeneous and heterogeneous agents. We briefly introduce these SMAC challenge tasks in Table 5.

Besides QMIX and QPLEX, we compare our methods with a state-of-the-art baseline QTRAN. Sufficient experiments show that our methods can achieve competitive performance with these SOTA baseline approaches.

### D.2  Empirical Results

We follow the original experimental settings without any hyper-parameter tuning. Fig 1 plots the median training performance of the test win rate across all the six scenarios. From the result, we can see that `QSCAN` and `QPAIR` are superior to QMIX and QTRAN in most scenarios. Moreover, our methods reveal a competitive performance compared with the SOTA baseline QPLEX, which uses the joint action in the mixing function.

## E  Additional Experiments

### E.1  Additional Results for Predator-Prey

QPLEX fails to learn a policy which achieves positive rewards in 8 vs 8 and 10 vs 10, although it finds some effective policies in 6 vs 6. The large number of joint actions and the lack of the

Table 2: The network configurations of `QPAIR`'s architecture.

| QPAIR's architecuture configurations | For matrix game | For others |
| --- | --- | --- |
| The number of MLP layers in Transformation | 3 | 1 |
| The number of heads in the multi-head attention | 10 | 4 |
| Unit number in middle layers | 64 | $\emptyset$ |
| Activation in the middle layers | ReLU | $\emptyset$ |
| Activation in the last layer of multi-head attention | Sigmoid | Sigmoid |
| Activation in the last layer of $\lambda, w$ | ReLU | ReLU |

Table 3: The network configurations of `QSCAN`'s architecture.

| QSCAN's architecuture configurations | For matrix game | For others |
| --- | --- | --- |
| Embedding dimensions of input | 64 | 64 |
| The number of heads in the self-attention | 10 | 4 |
| Dimension of hidden layer in self-attention | 64 | 64 |

Table 4: Default hyper-parameters.

| Hyper-parameter | Value | Description |
|---|---|---|
| n_runs | 6 | Number of training runs for each task |
| learning_rate | 0.0005 | Learning rate used by Adam optimizer |
| replay_buffer_size | 5000 | Maximum number of samples to store in memory |
| minibatch_size | 32 | Number of samples to use for each update |
| n_test_episode | 32 | Number of episodes for evaluation |
| discount_factor | 0.99 | Importance of future rewards |
| target_update_period | 200 | Target network update period to track learned network |

Table 5: SMAC challenges.

| Map Name | Ally Units | Enemy Units |
|---|---|---|
| 2s_vs_1sc | 2 Stalkers | 1 Spine Crawler |
| 2s3z | 2 Stalkers, 3 Zealots | 2 Stalkers, 3 Zealots |
| 3s5z | 3 Stalkers, 5 Zealots | 3 Stalkers, 5 Zealots |
| 1c3s5z | 1 Colossus, 3 Stalkers, 5 Zealots | 1 Colossus, 3 Stalkers, 5 Zealots |
| 2c_vs_64zg | 2 Colossi | 64 Zerglings |
| 5m_vs_6m | 5 Marines | 6 Marines |
| 27m_vs_30m (Super-Hard) | 27 Marines | 30 Marines |

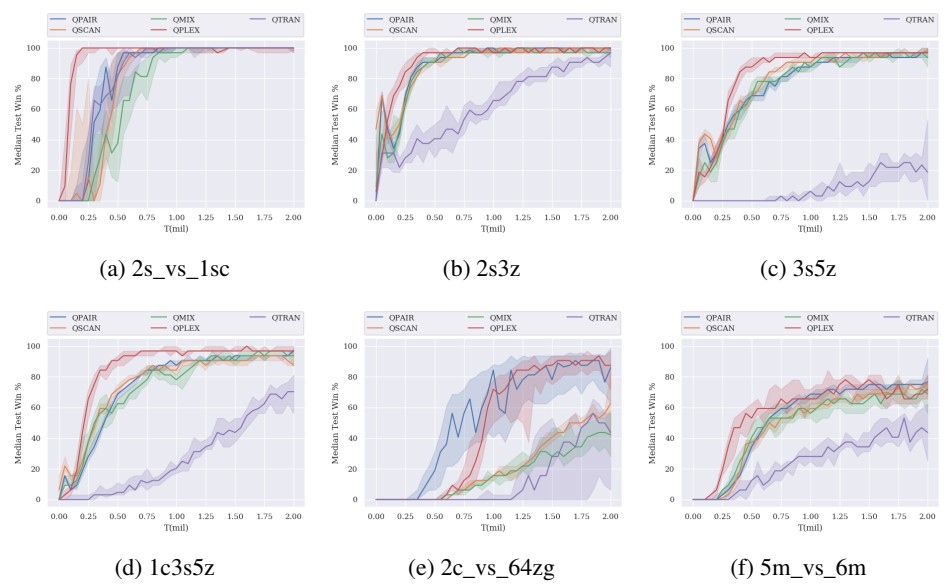

Figure 1: Median performance of the test win percentage in SMAC.

generalisation prevent QPLEX from extracting the correct coordination of "capture" actions. The performance of QMIX improves in 8 vs 8 and 10 vs 10. With the agent number increasing, the probability of successful capture increases. As a result, the issue of relative overgeneralization eases and QMIX finally realizes that "capture" is better than doing nothing. However, as shown in Fig 2b, the performance of QMIX drops finally. QMIX still suffers the issue of relative overgeneralization. As for QPAIR, it performs better than QMIX in 6 vs 6, while it behaves worse in 8 vs 8 and 10 vs 10. One reason is that QPAIR is forced to learn only the pairwise coordination without learning each individual's local information. Although this local information can be encoded into the pairwise coordination, the enforcement constraints hurts the stability of the optimization process. Thing will aggravate even further when the number of agent increases. For QSCAN, it outperforms for all three scenarios due to its adaptive balance of the pairwise coordination and each individual's local information.

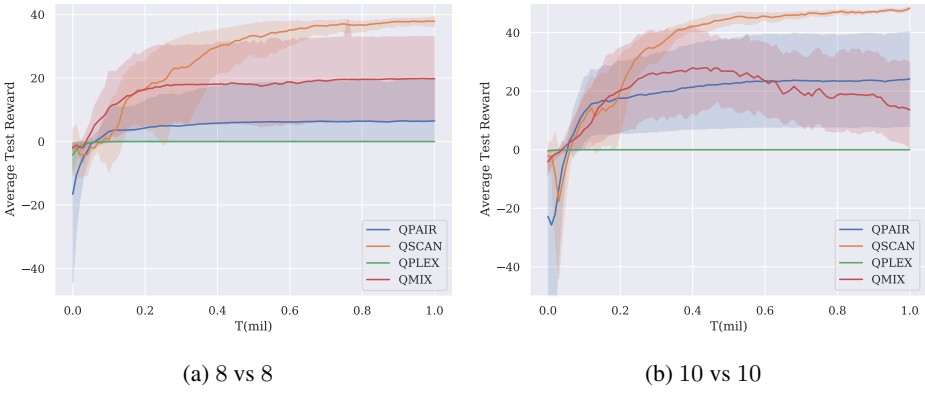

(a) 8 vs 8               (b) 10 vs 10

Figure 2: Learning curves in two additional Predator-Prey tasks.

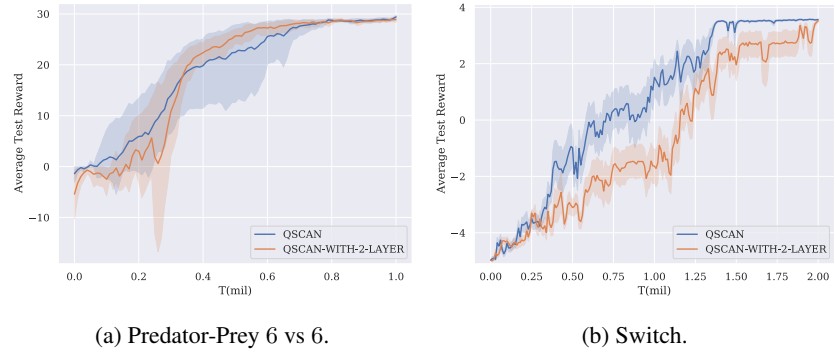

(a) Predator-Prey 6 vs 6.        (b) Switch.

Figure 3: Learning curves of `QSCAN` with different numbers of self-attention layers.

## E.2 Ablation Study on `QSCAN`'s Layers

We examine our approach `QSCAN` with different numbers of self-attention layers in two different tasks: Predator-Prey 6 vs 6 and Switch. `QSCAN-WITH-2-LAYER` has 2 self-attention layers and emphasizes the sub-team coordination with size at most 4.

**Ablation experiments for Predator-Prey 6 vs 6.**

Our ablation experiments in Fig 3a show that the sub-team coordination with size at most 4 is more suitable for this task. `QSCAN-WITH-2-LAYER` achieves better performance than `QSCAN` with minor variance. Although the pairwise coordination is enough for accomplishing this 6 vs 6 task, the sub-team coordination with a larger size can do better. The pairwise coordination can only organize one "capture" while the sub-team coordination with size 4 can schedule more "capture"s effectively.

**Ablation experiments for Switch.**

Our ablation experiments in Fig 3b show that the sub-team coordination with size at most 4 performs worse than the sub-team coordination with size at most 2 in this task, whereas they achieve similar final performance. Since the one-agent wide corridor has only two entrances and there are precisely two agents at each side of the corridor, the pairwise coordination is enough for the coordination at each side and the organization of the side-by-side passing process. The sub-team coordination with size at most 4 enlarges the model complexity and makes the learning process more complicated than that of the pairwise coordination.

**Summary**

The two different ablation experiments above show that encoding the sub-team coordination with a suitable size into the network would bring about higher sample efficiency and lower variance in the training process. With proper prior knowledge of a task, we can choose a suitable coordination level.

| Predator-Prey Task | Approach | # Parameters | Final Score |
|---|---|---|---|
| 8 vs 8 | QPAIR | 51.720K | 6.50 |
| | QSCAN | 417.745K | * 37.86 |
| | QPLEX | 61.572K | 0.00 |
| | QPLEX-Large | 784.058K | 0.00 |
| 10 vs 10 | QPAIR | 54.404K | 24.16 |
| | QSCAN | 418.517K | * 48.32 |
| | QPLEX | 67.512K | 0.00 |
| | QPLEX-Large | 794.598K | 0.00 |

Table 6: Ablation on Number of Parameters

### E.3  Additional Comparison with QPLEX

The self-attention module of QSCAN uses a large number of parameters. For a fair comparison, we additionally run a larger QPLEX version (QPLEX-Large) that can be found in origin QPLEX code. Table 6 shows the numbers of parameter and the final score of each approach for two Predator-Prey tasks (8 vs 8, and 10 vs 10).

For each tasks, QPAIR uses similar number of parameters with the origin QPLEX, and QSCAN uses similar number of parameters with the larger QPLEX. Both of our approaches outperform the corresponding version of QPLEX.

### E.4  Additional Results in a Super-Hard SMAC Task

We additionally compare our approach QSCAN with the SOTA baselines QPLEX and QMIX in a super-hard SMAC task, 27m_vs_30m. Fig 4 shows that QSCAN strikes a better trade off between the complexity of the hypothesis space and the capacity of the representation, and outperforms two baselines.

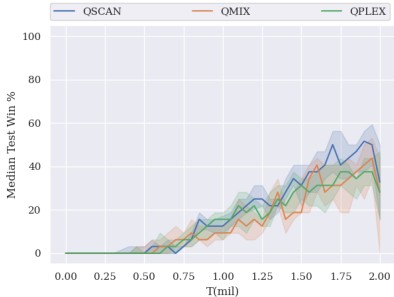

Figure 4: Median performance of the test win percentage on 27m_vs_30m