# OpenReview forum: "Multiagent Q-learning with Sub-Team Coordination"
_NeurIPS.cc/2022/Conference — NeurIPS 2022 Accept_

### Official Review · Reviewer_MzdJ · 2022-06-30

**Rating:** 6
**Confidence:** 4
**Soundness:** 3 good
**Presentation:** 2 fair
**Contribution:** 3 good

**Summary:**

The paper proposes a hierarchical approach to value function factorization called QSCAN, where the multi-agent team is divided into sub-teams of fixed size, which solve different sub-tasks, in order to exploit local coordination. The architecture of QSCAN is based on QPLEX, where a sub-team coordination module determines the contribution weights of each agent for all potential sub-teams. The paper presents two instantiations of QSCAN called QPAIR, enumerating all sub-teams of size 2, and QSCAN, using self-attention on the agents themselves, which are evaluated against QMIX and QPLEX in various benchmark domains.

**Questions:**

- Line 225: Is the importance weight $g^{\textit{ST}}_i$ assumed to be non-negative?
- How many training runs were conducted per task? I could not find any information in the paper or appendix.

**Limitations:**

The provided checklist refers to Section 6, but I did not find any explicit limitations. There is a statement about a challenge of designing more effective hierarchical structures but the actual limitation is unclear to me, given that the results are mostly in favor of QPAIR/QSCAN anyway.

Potential negative societal impact is not discussed.

**Strengths And Weaknesses:**

- **Originality**: A similar idea about sub-team based hierarchical value factorization was published in NeurIPS 2021 [1]. However, the introduced coordination hierarchy with QMIX and QPLEX as special cases is interesting and novel in my view.
- **Quality**: The paper seems technically sound and the performance of QSCAN is supported in the experiments. The number of agents in all benchmarks is rather small, which would make an evaluation with $k > 2$ teams feasible. Regarding the claim about focusing on local coordination, I recommend to evaluate in SMAC scenarios with > 10 agents like ``bane_vs_bane`` rather than ``2c_vs_64zg``, where the single sub-team is obvious.
- **Clarity**: The approach is well explained. I did not check the attention heat-maps in Figure 6 because the plots are too small, making it very hard to follow the descriptions in the text.
- **Significance**: QPAIR and QSCAN seem to clearly outperform QMIX and QPLEX in most settings. However, I could not find any information about the number of training runs. The error/deviation bars in Table 2 regarding the SMAC results are missing. Thus, I cannot assess the significance of the results.

Minor comments:
- Line 72: QPAIR is mentioned for the first time without further description, which is confusing. The QSCAN variant should be named differently from its actual framework as it is not obvious that QSCAN is used in different contexts in the same parapgraph.
- Line 91: $r: \mathcal{S} \times \mathcal{A} \rightarrow \mathbb{R}$ denotes __the__ global reward function
- The text and labels on all figures are very small and hard to read when printed (the white texts in Figure 4 are completely unreadable)
- Line 283: „several state-of-the-art MARL approaches“ are just two approaches: QMIX and QPLEX

[1] Phan et al., "VAST: Value Function Factorization with Variable Agent Sub-Teams", NeurIPS 2021

---

> ### Author Response · Authors · 2022-08-02
> **Response to Reviewer MzdJ**
>
> We thank the reviewer for the review. We are encouraged that the reviewer found our coordination hierarchy interesting.
>
> We’ll directly incorporate some of the recommendations in our revision. Here are our responses to some of the questions:
>
> **Q1:** Line 225: Is the importance weight $g_{i}^{ST}$ assumed to be non-negative?
>
> **A1:** No, $g_{i}^{ST}$ could be negative. Agent $i$'s action $a_{i}$ might contribute to a sub-team $ST$, while such action could have negative impact on another sub-team $ST'$. In fact, only the weights $\lambda_{i}$ should be non-negative.
>
> **Q2:** How many training runs were conducted per task? I could not find any information in the paper or appendix.
>
> **A2:** We take $6$ training runs for each task. Sorry for omitting this information. It will be added to our revised version.
>
> **Reviewer:** I recommend to evaluate in SMAC scenarios with > 10 agents like `bane_vs_bane` rather than `2c_vs_64zg`, where the single sub-team is obvious.
>
> **Response:** Thanks for the recommendation. In Appendix E.4, we have provided the empirical result for `27m_vs_30m`, which is a superhard SMAC scenario with > 10 agents. We will consider a comparison for other such scenarios like `bane_vs_bane` if our computational resources are sufficient.
>
> **Reviewer:** The error/deviation bars in Table 2 regarding the SMAC results are missing.
>
> **Response:** Thanks for the recommendation. The error bars can be found in the corresponding subfigures of Fig. 1 in Appendix E.4.
>
> **Reviewer:** There is ... the results are mostly in favor of QPAIR/QSCAN anyway. Potential negative societal impact is not discussed.
>
> **Response:** The **challenge** is to consider other sub-team factorization frameworks beyond the factorization on advantages, which could be completely different from QPAIR/QSCAN. Considering the policy information rather than the action information could be a possible direction. The general societal impact could be found in [1]. The potential negative societal impact of QSCAN could be the fairness problem: A concise factorization could increase the global value while some agents will do more (dangerous) things than others if they do these better, which could be a crucial fairness problem.
>
> > [1] Whittlestone, Jess, Kai Arulkumaran, and Matthew Crosby. "The societal implications of deep reinforcement learning." Journal of Artificial Intelligence Research 70 (2021): 1003-1030. https://www.jair.org/index.php/jair/article/download/12360/26667

---

> > ### Comment · Reviewer_MzdJ · 2022-08-05
> > **Response to authors**
> >
> > Thank you for addressing my minor comments.
> >
> > I still have some unaddressed major concerns:
> > - As reviewer 3a1Z and I pointed out, there is a missing comparison with VAST (see [1] in the original review). Given that it is a highly related approach, merely citing it in the appendix is not sufficient. It needs to be discussed in the main paper in a proper related work section.
> > - An evaluation with $K > 2$ agents is appreciated in the smaller domains, where sub-teams can be easily enumerated, e.g., Predator-Prey with 6 agents (with a maximum of 20 possibilities if $K = 3$). Otherwise the impression of the whole concept degenerates to a simple special case.
> >
> > Regarding the challenge: I do understand the formulated challenge. But I do not understand how it relates to the **limitations** of the presented approach (which is important as the checklist asks for current limitations and not for future challenges). For example: What are the potential consequences of leaving that challenge open?

---

> > > ### Author Response · Authors · 2022-08-07
> > > **Response to unaddressed concerns**
> > >
> > > Thanks so much for the reply. We really appreciate it and will use it to revise our paper.
> > >
> > > **About comparison with VAST:** We have tested our approach QSCAN in some VAST paper's environments. The empirical results show that our method is comparable with VAST in these environments. You can check the general response for detail.
> > >
> > > **About the evaluation with $K>2$:** An ablation study about $K$ for QSCAN can be found in **Appendix E.2**. We ablate the sub-team size through the number of attention layers: one layer for size $2$, and two layers for size $4$. The empirical results show that the sub-team size influences the training result: In Predator-Prey, QSCAN with two attention layers performs slightly better, while it performs worse in Switch. For QPAIR, we do not implement the simple extension with $K>2$ as it has limited usage in a bit large multiagent systems. For example, in the SMAC scenario `27m_vs_30m`, the number of size-$3$ sub-teams is about $3000$, and the time cost might not be acceptable. Due to limited computing resources, we are afraid we have no time to add the QPAIR results with $K=3$. We will add these experimental results in the future.
> > >
> > > **About the challenge:** Our current approach's issue could be simplifying the sub-team representation. Our framework attempts to cover most sub-team coordination patterns, but this may increase the computation cost. As we mentioned in **Sec. 6**, there has not been a universal organizational paradigm suitable for most tasks. For example, consider a company developing two unrelated software, e.g., one software for the bank system and one software for PC users. Each software needs about half of the software engineers in the company. However, each engineer can develop one software due to the business security issue. To represent such kinds of organizations, QPAIR needs to enumerate $\binom{n}{\frac{n}{2}}$ sub-teams, QSCAN needs about $O(\log n)$ attention layers for computing, but VAST (or some methods representing coalitions) needs only constant number of layers.
> > > We will modify our representation in **Sec. 6** to show this limitation more clearly.

---

### Official Review · Reviewer_wAct · 2022-07-11

**Rating:** 6
**Confidence:** 3
**Soundness:** 3 good
**Presentation:** 4 excellent
**Contribution:** 3 good

**Summary:**

In this paper, the authors proposed a method called QSCAN to perform value-based decomposition on a subset of agents in multi-agent reinforcement learning. For the given task, the idea is to arrive at weights by which Q-functions are scaled to get an idea of sub-team coordination. The proposed technique is tested on different environments and obtained good performance when compared with different existing methods.

**Questions:**

Please refer to my points in the weakness of the paper.

**Limitations:**

There are no limitations addressed in the proposed approach. There are some typos in the paper which the authors need to correct.

**Strengths And Weaknesses:**

The paper is structured well and written well. The paper deals with an interesting problem that I believe no one is dealt with before. The problem is of good importance in industries where there will be multiple tasks and only a subset of agents participate in a single task. However, below are some of the points which I think may improve the paper.

1. How the weights $\lambda_i$ are learned over time. Do authors use the regular experience replay buffer for this case? I suggest incorporating more details here.

2. What about the scalability of the proposed approach? I see the computational complexity is so enormous (especially when the number of combinations to try out is huge).

3. I am confused on the execution part other proposed approach. Are weights calculated for every time step?

---

> ### Author Response · Authors · 2022-08-02
> **Response to Reviewer wAct**
>
> We thank the reviewer for the encouraging review. We appreciate that the reviewer found our paper well-written and our problem interesting. We'll try to address the questions here:
>
> **Q1:** How the weights λ are learned over time. Do authors use the regular experience replay buffer for this case? I suggest incorporating more details here.
>
> **A1:** (1) The weights $\lambda_{i}$ are learned by the end-to-end training. We do not design a specific training period for  $\lambda_{i}$. (2) Yes, we use the replay buffer with size $5000$ as QMIX and QPLEX. This parameter can be found in Table 4 in Appendix.
>
> **Q2:** What about the scalability of the proposed approach? I see the computational complexity is so enormous (especially when the number of combinations to try out is huge).
>
> **A2:** As we mentioned in **Sec 5.**, the simple extension of QPAIR needs exponential computations, while QSCAN is designed to ease this issue. For a team of $n$ agent and sub-teams with size $k$, the extension of QPAIR needs to compute all $O(n^k)$ sub-teams, while QSCAN employs about $O(\log k)$ attention layers to approximate the results. We believe the complexity for QPAIR is $O(n^{k+1})$ and that for QSCAN is $O(n\log k)$, which means QSCAN is more scalable than QPAIR.
>
> **Q3:** I am confused on the execution part other proposed approach. Are weights calculated for every time step?
>
> **A3:** Yes, the weights should be recalculated at each time step. One simple explanation is that, the reward functions at different time steps are different matrix games, so the optimal sub-team factorizations should be different.

---

### Official Review · Reviewer_f854 · 2022-07-17

**Rating:** 6
**Confidence:** 4
**Soundness:** 3 good
**Presentation:** 2 fair
**Contribution:** 3 good

**Summary:**

Authors present QSCAN, an attention-based, sub-team-level organization, task allocation, and credit assignment algorithm to outperform certain special cases of MARL algorithms that fall under the sub-team organization, namely, QMIX (k=1) and QPLEX (k=n). QSCAN is evaluated in three environments against other factorization methods and shows better performance consistently.

**Questions:**

Q1: How important is the addition of self-attention? It would be nice if the authors could share some ablated evaluation results, which are probably available if they begun initially without attention.

Q2: What is the effect of altering k but for the same evaluation setting? What is the (expected) effect of forming different-size sub-teams?

**Limitations:**

Please have a look into the effects of different k as well as the effects of attention. Other than that, the limitations of QSCAN have already been clearly stated in Section 6.

**Strengths And Weaknesses:**

QSCAN ties in well with the literature, and the evaluation is carried out under varied settings.

The paper's weakness lies in its lack of proof for the initial claim of more effective factorization and faster learning. Sub-team organization and attention are additions to the computational complexity, so the number of steps in Fig. 5 cannot vouch for "faster" learning. For the SMAC evaluation, even the number of steps data is omitted for reasons unknown. On a separate note, some explanation on role assignment line of MARL research (e.g., RODE) could make the analyses more comprehensive, as role assignments (with no explicit team formation) and sub-team formation (with no explicit role definition) could be positioned interestingly as duals.

---

> ### Author Response · Authors · 2022-08-02
> **Response to Reviewer f854**
>
> We thank the reviewer for the comments and questions.
> We are happy to hear that the reviewer found our work "tites in well with the literature".
> We will try to address the concerns/questions:
>
> **Q1:** How important is the addition of self-attention? It would be nice if the authors could share some ablated evaluation results, which are probably available if they begun initially without attention.
>
> **A1:** As we mentioned in **Sec. 4**, we employ the self-attention mechanism in QSCAN to deal with the exponential enumeration issue caused by the simple extension of QPAIR. (1) If the review means the ablation of the enumeration and the attention, that is the comparison between QPAIR and QSCAN. (2) If the review means the ablation with/without the self-attention part for QSCAN, we should emphasize that QSCAN might not be able to represent the coordination patterns of sub-teams without the self-attention part.
>
> Could the reviewer specify how to *initially without attention*? We would be glad to add some ablation results in the final version of the paper.
>
> **Q2:** What is the effect of altering k but for the same evaluation setting? What is the (expected) effect of forming different-size sub-teams?
>
> **A2:** In general, altering $k$ (the sub-team size) would enrich or impoverish the representability of the sub-team coordination. Theoretically, the larger $k$ is, the more precisely the joint value function is approximated, the more complex the function space is, and the harder the solution is to optimize. An ablation study about $k$ for QSCAN can be found in **Appendix E.2**. As one attention layer aggregates the pairwise information of its inputs and two layers aggregate the quadruple-wise (pairwise of the pairwise) information, we ablate the sub-team size through the number of attention layers in QSCAN: one layer for sub-teams of size $2$; two layers sub-teams of size $4$. The empirical results show that the sub-team size influences the training result: In Predator-Prey, QSCAN with two attention layers performs slightly better, while it performs worse in Switch. The empirical results indicate that a suitable sub-team size helps to learn a better solution.
>
> **Reviewer:** The paper's weakness lies in its lack of proof for the initial claim of more effective factorization and faster learning.
>
> **Response:** We agree that we do not provide formal proof of the efficiency of factorization and learning. This initial claim is supported by several insights: (1) The optimal approximation won't get worse when increasing the representational capacity. (2) The optimal approximation will become harder to search when the representational capacity increases. (3) The coordination hierarchy encompasses the full spectrum of the sub-team coordination level, where QMIX and QPLEX are located at the respective extremes. (4) QMIX lies in the most inner circle so that it has a coarse-grained approximation with the simplest function class. (5) QPLEX lies in the most outer circle so that it has the most accurate approximation with the most complex function class. (6) When considering other middle circles, e.g., QSCAN$_k$, they take a tradeoff between the accuracy of the approximation and the complexity of function classes.

---

### Official Review · Reviewer_3a1Z · 2022-07-22

**Rating:** 5
**Confidence:** 3
**Soundness:** 3 good
**Presentation:** 2 fair
**Contribution:** 3 good

**Summary:**

This paper proposes QSCAN, which considers local coordination within a subset of the agents and dynamically solves each task by sub-team coordination. In order to incorporate sub-team coordination, QSCAN extends the monotonic mixing network in QPLEX and achieves the full representational capacity of IGM. The experimental results showed that QSCAN can outperform QPLEX and QMIX in a number of benchmarks, especially Predator-Prey and Switch4.


**Questions:**

1. What is the main benefit of using (1) sub-team coordination in general and (2) QSCAN in particular?
2. What is the main difference between QSCAN and VAST ?
3. How does QSCAN overcome the relative overgeneralization mentioned as an issue for QMIX?
4. Right before Proposition 1, what is the meaning of “satisfy the IGM Condition consistently during the course of training”?
5. Can we really say QSCAN significantly outperforms the benchmarks in the matrix game? Also, what do you mean by poor generalization for QPLEX in 5.1? QPAIR seems to have the same performance as QPLEX in the Matrix Game, around a reward value of 22. Thus, I think it would be misleading to say pairwise coordination patterns provide better generalization. Also, I am not convinced that the reward difference between 22 (QPAIR/QPLEX) and 23 (QSCAN) is significant. Is there a crucial point worth mentioning for this gap?
6. In the matrix game, what is the k used for QSCAN? Why is it able to perform (slightly) better than QPLEX/QPAIR? If the grouping is 2 sub-teams with 2 agents and 1 agent for QSCAN, are there some groups (e.g. player 1-2, 3) that can do well while other groupings don’t?


**Limitations:**

The authors outline the limitations of the work in the Conclusion (Section 6). I agree that going beyond advantage factorization is one of the main limitations in this work because QSCAN as it stands is somewhat limited by IGM and QPLEX.


**Strengths And Weaknesses:**

Strengths
- The authors provide a good motivation for coordinating sub-teams, which can be beneficial for tasks where agents should divide into groups and focus on solving each sub-task.
- The paper provides a nice sub-team generalization of some MARL algorithms where  QPLEX can be interpreted as QSCAN_n and QMIX as QSCAN_0. They provide a thorough theoretical analysis for various properties of QPLEX and QMIX such as the equivalence between QMIX and IGM within individual sub-team assignments (Proposition 1).
- The experimental results for Predator Prey and Switch4 are interesting. The environment states and attention maps given in Figure 6 provide interesting insight into the potential reasons for choosing a particular action. This can be viewed as one evidence for communication among agents via actions


Weaknesses:
- There is a lack of adequate comparisons with relevant benchmarks in sub-team coordination, especially VAST[18 (supp.)], which also proposes variable sub-teams beyond pairwise coordination and retains the IGM consistency. Sub-team coordination itself is not a novel concept, and I believe it is necessary to emphasize the differences between QSCAN, VAST [18], and other sub-team formulation algorithms (if any), both qualitatively and quantitatively as benchmarks.  VAST [18] is also working in cooperative MARL so it could be misleading to categorize them in the same group as the ones using coalitions and Shapley values.
- Related to the first point, the paper has a brief discussion of GNN-based approaches which are a different approach with similar motivations. The authors write in the supplementary material A that with GNNs, “it is difficult for them to extract hierarchical coordination patterns.” However, this claim was not verified in the experiments.
- With the lack of benchmarks against similar sub-team formulation algorithms and GNN-based ones, it is hard to agree with the statement that QSCAN “significantly outperforms” other benchmarks in the predator-prey and Switch4 tasks. Moreover, QSCAN only performs on-par with QPLEX on the SMAC tasks, and the improvement over QPLEX in the matrix game also appears marginal. So overall, the experimental results are not entirely convincing.
- There is mention of “relative overgeneralization”, “credit assignment”, “faster learning” scattered around the paper, but it is not clear what the main problem is that QSCAN aims to solve. I understand the intuitive motivation for sub-team formulation, but there is no characterization of the types of tasks in which the proposed algorithm is beneficial.
- Related to the previous point, the illustrative example given in Figure 1 is confusing since it only shows an example of when a sub-team is sufficient for solving the task. On the other hand, the experimental results seem to suggest that the main point of using sub-team formulation is to have a more flexible algorithm which can solve tasks which QPLEX/QMIX cannot, i.e. tasks in which sub-team formulation is necessary. Again, the main argument of the paper is ambiguous, especially with respect to why the extension of QPLEX is specifically the better method.
- VAST [1] shows that sub-team formulation can be useful for when the number of agents is sufficiently large. This is not necessarily a weakness of QSCAN but again, the main benefit of using QSCAN is not entirely clear. The benefit of using sub-team coordination (e.g. scalability to # of agents, fast convergence, etc.) and the benefit of using QSCAN in particular has to be properly addressed.
- The algorithm is derived from a fairly simple extension of QPLEX, where the architectural figures look almost identical.
- (Minor) Figure 6 is ineligible without zooming in.

EDIT: Much of my initial concerns in the review were clarified by the responses by the authors. The authors have shown that the flexible sub-team coordination is a novel problem which was not tackled in previous work. Furthermore, they have added additional experiments to show that the performance is comparable to VAST and scalable to a larger number of agents. To reflect this, I’ve raised my overall score (3→5) as well as the soundness (2→3) and contribution scores (2→3).

---

> ### Author Response · Authors · 2022-08-02
> **Response to Reviewer 3a1Z (1/3)**
>
> We thank the reviewer for the detailed feedback, and will use it to revise our paper.
> We are glad that the reviewer found our work "provides a good motivation for coordinating sub-teams" and "a nice generalization of MARL algorithms" and considered our experimental results "interesting".
> We are trying to address your questions here:
>
> **Q1:** What is the main benefit of using (1) sub-team coordination in general and (2) QSCAN in particular?
>
> **A1:** (1) As mentioned in *a survey of multi-agent organizational paradigms* [1], the sub-team organization has these benefits:
> * It is a general organizational paradigm. Nearly any cooperative agent system has characteristics that are similar to this organization, if only implicitly. Therefore, the sub-team organization provides a general structure to model the coordination patterns in cooperative multiagent systems.
> * It can address the ramifications of inter-agent interactions. In *towards flexible teamwork* [2], this paradigm is used to provide the structure and coordination needed by agents to address interdependent goals in dynamic environments.
>
> (2) The QSCAN framework employs the sub-team paradigm to obtain a series of value factorization functions. Two major benefits of QSCAN's sub-team factorization are as follows.
> * Sub-team factorization can represent general coordination patterns in multiagent systems.
> * The flexible factorization will benefit several cooperative multiagent tasks, for example, the interdependent tasks.
>
> Since the first major benefit has been mentioned in (1), we will briefly explain why flexible factorization is important.
> Intuitively, when the representation capacity increases, the global value of the optimal approximation solution will increase while such optimal approximation will become harder to search.
> A suitable representation capacity can balance the approximate result and the searching cost.
>
> To explain the interdependent tasks and the importance of the sub-team factorization, we will provide a real-life scenario here. Considering software development in a company, each software engineer could involve in several projects about this software. A project team shares engineers with other teams and can be treated as an agent-sharing sub-team. It should be noticed that the outcomes of the project teams are interdependent. Evaluating the software via each engineer's work ignores the coordination within the projects (sub-teams), while evaluating for all engineers together ignores the contribution of each project (sub-team) and makes things complex. Sub-team factorization strikes a tradeoff between the coordination within projects (representation capacity of coordination patterns) and the simplicity of the evaluation (the complexity of function classes).
>
> Empirically, we evaluate our approaches QPAIR and QSCAN in several interdependent scenarios similar to the software development example, e.g., Predator-Prey, and the result shows the outperformance of QSCAN.
>
> >[1] Horling, Bryan, and Victor Lesser. "A survey of multi-agent organizational paradigms." The Knowledge engineering review 19.4 (2004): 281-316. http://mas.cs.umass.edu/Documents/bhorling/horling-paradigms.pdf
>
> >[2] Tambe, Milind. "Towards flexible teamwork." Journal of artificial intelligence research 7 (1997): 83-124. https://www.jair.org/index.php/jair/article/download/10193/24210/

---

> ### Author Response · Authors · 2022-08-02
> **Response to Reviewer 3a1Z (2/3)**
>
> **Q2:** What is the main difference between QSCAN and VAST?
>
> **A2:** In VAST, the entire team is divided into different agent groups. In other words, any two groups are independent and have no intersection (see Sec 4.1 in the VAST paper [3]).
> Thus, the group structure in VAST is a *partition* of the team, and every group behaves like a coalition.
> In QSCAN, however, a sub-team is just a *subset* of the entire team, and two different sub-teams could share some agents.
> These sharing agents could execute actions to influence different sub-teams simultaneously and thus establish connections among them.
> At this point, the groups of VAST form a floating structure while the sub-teams of QSCAN form a connected hypergraph where the nodes are agents and hyperedges are the sub-teams.
>
> We also want to illustrate the difference via an intuitive example.
> Roughly speaking, we can differentiate the factorization of VAST, QSCAN, and QPLEX by Fig. 1 in [DCG](http://proceedings.mlr.press/v119/boehmer20a/boehmer20a.pdf) paper [4]: VAST is somewhat like Fig. 1.(a) that each disjoint sub-team contributes independently to the entire team if we treat a sub-team as a super agent; QSCAN is like Fig. 1.(b) that some sub-team might share agents; QPLEX is like Fig. 1.\(c\) that only the entire team is considered.
>
> Specifically, VAST uses the time-variable coalition as its organization, which allows each agent to belong to exactly one group (same as one coalition) at one perdicular time step.
> QSCAN's factorization allows an agent to take part in several sub-teams even at one time step. When focusing on one particular time step (or matrix games), VAST's sub-teams are the same as coalitions (that's why we mention VAST in the coalition part in Appendix A).
> From this viewpoint, VAST's factorization is a subclass of QMIX's monotonic factorization, which is why we did not consider VAST as a factorization baseline in our paper.
>
> >[3] Phan, T., Ritz, F., Belzner, L., Altmann, P., Gabor, T., & Linnhoff-Popien, C. (2021). VAST: Value Function Factorization with Variable Agent Sub-Teams. Advances in Neural Information Processing Systems, 34, 24018-24032. https://openreview.net/pdf?id=hyJKKIhfxxT
>
> >[4] Böhmer, Wendelin, Vitaly Kurin, and Shimon Whiteson. "Deep coordination graphs." International Conference on Machine Learning. PMLR, 2020. http://proceedings.mlr.press/v119/boehmer20a/boehmer20a.pdf
>
> **Q3:** How does QSCAN overcome the relative overgeneralization mentioned as an issue for QMIX?
>
> **A3:** Theoretically, <b>only QSCAN$_n$</b> can overcome the relative overgeneralization issue since QSCAN$_n$ can represent any possible joint value function. For QSCAN$_k$ $(k<n)$, it can <b>ease</b> this issue as it can approximate a more precise factorization than QMIX via its higher representational capacity.
> It could distinguish more kinds of cooperative patterns to achieve this.
>
> **Q4:** Right before Proposition 1, what is the meaning of “satisfy the IGM Condition consistently during the course of training”?
>
> **A4:** Proposition 1 tries to describe a property of the mixing function. During the training phase, the individual value functions $Q_{i}$ vary to optimize the accumulated global reward. At the beginning (or even during the course of training), the optimal solution remains (partially) unknown and could be anything. To ensure the IGM property during the course of training, $f$ should satisfy the IGM property for any possible value functions $\{Q_{i}\}_{i=1}^{n}$.
>
> Mathemetically, it means that
>
> $\forall s, \forall i, \forall Q_{i}:S \times A_{i} \mapsto \mathbb{R}, $
>
> the mixing function $f$ satisfies
>
> $~~~\max_{\vec{{a}}}{Q_{tot}(s, \vec{a})}$
>
> $=\max_{\vec{{a}}}{f(s, \{Q_{i}(s, a_{i})\}_{i=1}^{n})}$
>
> $=f(s, \{\max_{a_{i}}{Q_{i}(s, a_{i})}\}_{i=1}^{n}).$

---

> > ### Comment · Reviewer_3a1Z · 2022-08-03
> > **Comparison with VAST**
> >
> > Thank you for the detailed response. Together with the illustrative example of the software engineering project, the reasoning for excluding VAST (and the novelty/intuitive appeal of QSCAN) is a little clearer. To summarize my understanding, the argument is that (1) VAST can be viewed as a special case of QMIX where each agent can only be assigned to one sub-team (2) From Proposition 1, IGM and the QMIX factorization is equivalent (3) Thus, QMIX is able to represent any coordination pattern within the subclass (including VAST).
> >
> > While I understand this line of argument now (although it would be hard to deduce from the paper alone), personally I find it still a little awkward to exclude VAST as a baseline. Superficially, it is the closest algorithm in terms of sub-team coordination/full IGM/Cooperative MARL. More importantly, the experimental results are a little less convincing without VAST as a baseline, even though theoretically QMIX should be enough. Otherwise, it seems that the benefit of flexible factorization (QSCAN) over coalitions (VAST) is less clear. This may especially be the case since VAST claims to outperform QMIX as well due to better credit assignment, albeit in different environments.
> >
> > Thus, if the time and computational resources permit, it would be better if there are further experimental results comparing QSCAN with VAST, at least for the ones excluding SCII.

---

> > > ### Author Response · Authors · 2022-08-07
> > > **Response to the experiments comparing with VAST**
> > >
> > > Many thanks for the reply. We really appreciate it and will make our presentation clearer and add these experimental results in our final version.
> > >
> > > We have compared our approach QSCAN with VAST in some environments. You can see the general response for detail.

---

> > > > ### Comment · Reviewer_3a1Z · 2022-08-08
> > > > **Thank you for adding additional experimental results.**
> > > >
> > > > Thank you for the additional results comparing VAST. Please see the updated scores and comments under EDIT.

---

> ### Author Response · Authors · 2022-08-02
> **Response to Reviewer 3a1Z (3/3)**
>
> **Q5:** Can we really say QSCAN significantly outperforms the benchmarks in the matrix game? Also, what do you mean by poor generalization for QPLEX in 5.1? QPAIR seems to have the same performance as QPLEX in the Matrix Game, around a reward value of 22. Thus, I think it would be misleading to say pairwise coordination patterns provide better generalization. Also, I am not convinced that the reward difference between 22 (QPAIR/QPLEX) and 23 (QSCAN) is significant. Is there a crucial point worth mentioning for this gap?
>
> **A5:** We agree that the rewards accrued by the two algorithms may look quite similar, although QSCAN obtains the optimal reward $23$ (at the top-lower-left corner of the matrix). However, it should be noticed that the sub-optimal solution (at the bottom-upper-right corner of the matrix) is $20$. For an average reward value of $22$ (actually, QPLEX obtains about $21.5$ at the last iteration), the sub-optimal solution is found with a probability of about $33\%$. From this viewpoint, we say QSCAN outperforms others significantly. We would add more explaination in our revised version.
>
> **Q6:** In the matrix game, what is the k used for QSCAN? Why is it able to perform (slightly) better than QPLEX/QPAIR? If the grouping is 2 sub-teams with 2 agents and 1 agent for QSCAN, are there some groups (e.g. player 1-2, 3) that can do well while other groupings don’t?
>
> **A6:** We would first clarify one point that, when grouping $3$ agents $\{1,2,3\}$ with $2$ sub-teams, the two sub-teams, not similar with coalitions, could be $\{1,2\}$ and $\{1,3\}$. (1) As we mentioned in **Sec. 5**, we use $k=2$ (sub-teams with size $2$). (2) The reason might be: QPLEX might trap in the sub-optimal solution as it considers the coordination of the entire team (with size $3$), and this coordination pattern is more complex and harder to optimize; QPAIR might not consider the individual (can be seen as sub-teams with size $1$) patterns well as it enforces to use the coordination patterns of sub-teams of exact size $2$, although the individual patterns can be mathematically represented by the coordination patterns used by QPAIR. QSCAN (3) Possibly not. Our framework consider the sub-teams with size $2$ rather than $2$ sub-teams. QSCAN/QPAIR considers the coordination patterns in sub-teams $\{1,2\}$, $\{1,3\}$, and $\{2,3\}$ simultaneously, rather than that in sub-teams $\{1,2\}$ and $\{3\}$.
>
> **Reviewer:** Related to the first point, the paper has a brief discussion of GNN-based approaches which are a different approach with similar motivations. The authors write in the supplementary material A that with GNNs, “it is difficult for them to extract hierarchical coordination patterns.” However, this claim was not verified in the experiments.
>
> **Response:** We agree with the reviewer that there could be some GNN-based approaches to extract hierarchical coordination patterns, and we would change the language in the supplementary material to remove this claim. How to extract these patterns well for GNN-based approaches is an interesting problem, and we will consider this problem in the future. In our view, the discussion and the claim are about the existing GNN-based value factorization frameworks. To the best of our knowledge, these frameworks inputs with only individual action values $\{Q_{i}\}_{i=1}^{n}$ and the state/history information. To ensure the IGM property, the mixing GNN satisfies the monotonic condition as QMIX (e.g., GraphMIX [5]). As we know that QMIX cannot extract hierarchical coordination patterns well (e.g., Matrix game in QTRAN paper [6] or Predator-Prey in DCG paper [4]), it is difficult for the existing frameworks to extract such patterns.
> We would explore the GNN-based architectures for extracting hierachical coordination patterns in the future.
>
> > [5] Naderializadeh, Navid, et al. "Graph convolutional value decomposition in multi-agent reinforcement learning." arXiv preprint arXiv:2010.04740 (2020). https://arxiv.org/pdf/2010.04740.pdf
>
> > [6] Son, Kyunghwan, et al. "Qtran: Learning to factorize with transformation for cooperative multi-agent reinforcement learning." International conference on machine learning. PMLR, 2019. http://proceedings.mlr.press/v97/son19a/son19a.pdf

---

### Author Response · Authors · 2022-08-07
**We conduct additional experiments for comparing our methods with VAST**

We thank the reviewers again for their insightful and thorough feedback. We notice that two reviewers point out that a comparison with VAST is needed for our paper. We will first briefly illustrate the differences between QSCAN and VAST. After that, we show several experimental results for comparing our methods with VAST:

- Results of QSCAN on VAST paper's environments.
- Result of VAST on SMAC.


## Key Differences Between QSCAN and VAST

Besides the differences in organizational paradigm, which we have explained in previous responses, we would like to point out that QSCAN and VAST follow *different learning frameworks*:
- VAST follows the **actor-critic** learning method (See Sec. 5.2 in VAST paper [1]).
- QSCAN and QPAIR are **multiagent q-learning** methods.

## Experimental Domains

1. For a fair comparison, we compare QSCAN and VAST via the same comparison method between VAST and QMIX in the VAST paper [1].
2. We notice that VAST did not run experiments on SMAC in its paper. As SMAC is a common-used cooperative multiagent benchmark, we compare VAST on some SMAC scenarios.

## Results of QSCAN on VAST Paper's Environments

Due to the limited time and computing source,
we can only test QSCAN on the environments *Warehouse-4*, *Battle-20*, and *GaussianSqueeze-200* in the VAST repo (https://github.com/thomyphan/scalable-marl).
Here are the results with $6$ runs for each environment:
|Environment|VAST Result|QSCAN Result|Final Result of QSCAN|
|-----------|-----------|-------|-----------|
|**Warehouse-4**|![Warehouse-4-VAST](https://i.imgur.com/k7THIoa.png#pic_right =200x)|![Warehouse-4](https://i.imgur.com/s36QUae.png#pic_right =200x)|6.64|
|**Battle-20**|![Battle-20-VAST](https://i.imgur.com/sQyxVoM.png#pic_right =200x)|![Battle-20](https://i.imgur.com/QDEo0sC.png#pic_right =200x)|11.51|
|**GaussianSqueeze-200**|![GuassianSqueeze-200-VAST](https://i.imgur.com/EFx92Di.png#pic_right =200x)|![GuassianSqueeze-200](https://i.imgur.com/EcT97ea.png#pic_right =200x)|413.94|

Compared with corresponding VAST's result, QSCAN is comparable with VAST in these environments and outperforms VAST on *Battle-20*.

## Result of VAST on SMAC

We test one of the VAST's implementations VAST-QMIX on a SMAC scenario `5m_vs_6m` with $4$ runs. Comparing the test reward of VAST-QMIX with that of QMIX, QMIX dominates VAST-QMIX (see the following picture).

![5m\_vs\_6m](https://i.imgur.com/KXpNioz.png =200x)

As we do not retain the test reward mean of QSCAN/QPAIR, we are not able to draw QSCAN/QPAIR in the picture above. However, by an indirect comparison through the test win rate (see Table 2 in **Sec. 5.4** or Fig. 1.f in **Appendix**), QSCAN/QPAIR outperforms QMIX, and thus outperforms VAST-QMIX.

We are still running some experiments of VAST on other SMAC scenarios. Once the experiments are accomplished, we will add them to the response.

## Conclusion

As the empirical results above show, QSCAN can achieve comparable results with VAST on the environment in VAST paper. And for the widely-used cooperative MARL benchmark, SMAC, VAST performs poorly. It could be the reason that several actor-critic methods, such as COMA [2], cannot perform as well as multiagent q-learning methods on SMAC. As QSCAN and QPAIR do not face this shortcoming, they achieve comparable results with current SOTA methods.

> [1] Phan, T., Ritz, F., Belzner, L., Altmann, P., Gabor, T., & Linnhoff-Popien, C. (2021). VAST: Value Function Factorization with Variable Agent Sub-Teams. Advances in Neural Information Processing Systems, 34, 24018-24032. https://openreview.net/pdf?id=hyJKKIhfxxT

> [2] Foerster, Jakob, et al. "Counterfactual multi-agent policy gradients." Proceedings of the AAAI conference on artificial intelligence. Vol. 32. No. 1. 2018. https://ojs.aaai.org/index.php/AAAI/article/view/11794

---

> ### Author Response · Authors · 2022-08-09
> **Additional results of VAST on SMAC**
>
> We have just finished experiments of VAST($\eta=0.5$) and VAST($\eta=0.25$) (as **Sec. 6.2** in VAST paper [1]) on SMAC [2] tasks `3m`, `8m`, `1c3s5z`, `3s5z`, `5m_vs_6m`. `3m` and `8m` are two easy tasks in SMAC, and we use them to examine whether VAST can solve SMAC-style tasks. Meanwhile, we use the results on one easy task (`1c3s5z`) and two hard tasks (`3s5z`, `5m_vs_6m`) to compare VAST with our approaches. Results of our approaches QPAIR and QSCAN on these three tasks can be found in Table 2 in **Sec. 5.4** or Fig. 1.f in **Appendix**.
>
> Here are the results:
>
> |Maps|Difficulty|VAST Result|
> |---------|------------|-------------|
> |`3m`|Easy|![vast_3m](https://i.imgur.com/sDTUTI4.png =200x)|
> |`8m`|Easy|![vast_8m](https://i.imgur.com/1F6hlxv.png =200x)|
> |`1c3s5z`|Easy|![vast_1c3s5z](https://i.imgur.com/KmTDvfC.png =200x)|
> |`3s5z`|Hard|![vast_3s5z](https://i.imgur.com/8D1FBbJ.png =200x)|
> |`5m_vs_6m`|Hard|![vast_5m_vs_6m](https://i.imgur.com/cRcCdzv.png =200x)|
>
> In all $5$ tasks, QMIX dominates two VAST implements, and VAST($\eta=0.5$) dominates VAST($\eta=0.25$). Unfortunately, even on `3m` (one of the easiest SMAC tasks), VAST can merely find a winning strategy. These results enhance our conclusion in the previous general response that VAST faces some shortcomings on SMAC tasks while our approaches do not.
>
> > [1] Phan, T., Ritz, F., Belzner, L., Altmann, P., Gabor, T., & Linnhoff-Popien, C. (2021). VAST: Value Function Factorization with Variable Agent Sub-Teams. Advances in Neural Information Processing Systems, 34, 24018-24032. https://openreview.net/pdf?id=hyJKKIhfxxT
>
> > [2] Samvelyan, Mikayel, et al. "The starcraft multi-agent challenge." arXiv preprint arXiv:1902.04043 (2019). https://arxiv.org/pdf/1902.04043

---

### Meta-Review · Area_Chair_tuA6 · 2022-08-26

**Recommendation:** Accept
**Confidence:** Certain

**Metareview:**

This paper presents QSCAN, a hierarchical approach to value function decomposition where the agents are grouped into sub-teams to solve different sub-tasks via local coordination. QSCAN extends the monotonic mixing network in QPLEX in order to represent sub-team coordination, and shown to outperform QPLEX and QMIX in a number of benchmark tasks. Reviewers were all in favor of accepting the paper based on the sound contribution of the proposed approach. However, figures are illegible due to font sizes. Please fix the figures in the final version of the paper.

**Award:**

No

---

### Decision · Program_Chairs · 2022-09-14

Accept